# CONSTRAINT-AWARE REWARD RELABELING FOR OFFLINE SAFE REINFORCEMENT LEARNING

## ABSTRACT

Offline safe reinforcement learning (OSRL) considers the problem of learning reward-maximizing policies for a pre-defined cost constraint from a fixed dataset. This paper proposes a simple and effective approach referred to as Constraint-aware Reward (Re)Labeling (CARL), that can be wrapped around existing offline RL algorithms. CARL is an iterative approach that alternates between two steps for each sampled batch of data to ensure state-action-wise safety constraints. First, update cost evaluation function using an off-policy evaluation procedure. Second, update policy using relabeled rewards (assign large penalty) for state-action pairs which are detected unsafe based on cost estimates. CARL is a minimalist approach, doesn't introduce any additional task-specific hyperparameters, and allows us to leverage strong off-the-shelf offline RL algorithms to solve OSRL problems. Experimental results on the DSRL benchmark tasks demonstrate that CARL reliably enforces safety constraints under small cost budgets, while achieving high rewards. The code is available at `https://anonymous.4open.science/r/CARL-6F11`.

## 1 INTRODUCTION

Reinforcement learning (RL) has achieved remarkable success across diverse domains, from game playing (Silver et al., 2016) to robotic control (Siekmann et al., 2021). However, in safety-critical applications such as healthcare, autonomous systems, and industrial control, online exploration poses unacceptable risks. Offline reinforcement learning addresses this limitation by learning policies exclusively from pre-collected datasets without any additional interaction with the environment (Levine et al., 2020). However, in safety-critical domains, it is insufficient to focus on maximizing rewards alone and policies must also respect explicit safety constraints during deployment.

This double requirement motivates offline safe reinforcement learning (OSRL), where agents must simultaneously maximize expected returns while satisfying user-specified cost budgets or safety constraints. OSRL inherits fundamental challenges from both offline RL and safe RL: handling distributional shift from fixed datasets, and ensuring that the policy behavior remains within safety constraints after deployment. This problem becomes especially challenging under tight cost budgets, where even small cost constraint violations may be unacceptable.

As explained in the related work section, prior approaches for OSRL typically employ constrained optimization frameworks. These approaches rely on dual-gradient updates, Lagrangian multipliers, or policy regularization techniques (Polosky et al., 2022; Lee et al., 2022; Xu et al., 2022b). While theoretically principled, such methods introduce substantial algorithmic complexity. In practice, they can also be sensitive to hyperparameter configurations and require careful tuning to avoid training instability or collapsing to overly conservative solutions when the cost budget is small (Zheng et al., 2024).

This paper develops a novel approach referred to as *Constraint-aware Reward (Re)Labeling (CARL)* to solve OSRL problems. There are two key innovations behind CARL. First, we formulate an unconstrained policy optimization problem to enforce state-action-wise safety constraints. This formulation naturally motivates an iterative policy improvement algorithm that doesn't require tuning Lagrange multipliers. Second, we develop a simple iterative method that can be wrapped around offline RL algorithms with batch updates. It alternates between two steps for each sampled batch of data: update cost evaluation function using an off-policy evaluation procedure and update policy

using relabeled rewards (assigns a large penalty) for state-action pairs whose cost-to-go estimates violate cost constraint. CARL is a minimalist method requiring no additional hyperparameters, leveraging off-the-shelf offline RL algorithms to effectively address OSRL problems.

The main findings from our evaluation of CARL on DSRL benchmark tasks (Liu et al., 2024) are as follows. First, CARL produces safe policies with high returns on most tasks and outperforms prior methods on a greater number of tasks. Second, CARL achieves excellent performance for the challenging setting of small cost budgets. Third, when trained only on unsafe trajectories, CARL remarkably learns safe policies. Finally, CARL achieves good performance with different offline RL algorithms.

**Contributions.** The key contribution of this paper is the development and evaluation of the Constraint-aware Reward Relabeling approach for OSRL problems. Specific contributions include:

- Formulation of an unconstrained optimization problem for state-action-wise safety constraints.
- Development of the iterative constraint-aware reward relabeling (CARL) method that can be wrapped around existing offline RL algorithms.
- Experimental evaluation of the proposed CARL algorithm on DSRL benchmark tasks to demonstrate its strong effectiveness over state-of-the-art methods.

## 2 PROBLEM SETUP

We consider the problem of offline reinforcement learning under safety constraints, modeled using the Constrained Markov Decision Process (CMDP) framework. A CMDP is defined by the tuple $(\mathcal{S}, \mathcal{A}, P, r, c, \gamma, \mu_0)$, where $\mathcal{S}$ and $\mathcal{A}$ denote the state and action spaces, respectively; $P : \mathcal{S} \times \mathcal{A} \times \mathcal{S} \to [0, 1]$ represents the unknown stochastic transition dynamics; $r : \mathcal{S} \times \mathcal{A} \to \mathbb{R}$ is the reward function; $c : \mathcal{S} \times \mathcal{A} \to [0, C_{\max}]$ is a non-negative cost function; $\gamma \in (0, 1)$ is the discount factor; and $\mu_0$ is the initial state distribution.

Let $\pi : \mathcal{S} \to \mathcal{A}$ represent a policy that maps states to actions. Given $\pi$ and $\mu_0$, we can define a distribution over trajectories $\tau = \{(s_t, a_t, r_t, c_t)\}_{t=1}^{T}$ generated by rolling out $\pi$ from initial states drawn from $\mu_0$. We define

$$V_r^{\pi}(s) = \mathbb{E}_{\tau \sim \pi}\left[\sum_{t=1}^{T} \gamma^t r_t \mid s_1 = s\right], \qquad Q_r^{\pi}(s, a) = \mathbb{E}_{\tau \sim \pi}[V_r^{\pi}(s') \mid s_1 = s, a_1 = a]$$

to be the standard reward state- and action-value functions. Similarly $V_c^{\pi}(s)$ and $Q_c^{\pi}(s, a)$ denote the cost state- and ation-value functions respectively. In the offline setting, the agent has access only to a static dataset $\mathcal{D} = \{(s_i, a_i, r_i, c_i, s_i')\}_{i=1}^{n}$ collected from one or more unknown behavior policies, without further interaction with the environment.

The goal of offline safe RL (OSRL) is to learn a policy from the given offline dataset $\mathcal{D}$ that maximizes the expected return while satisfying a given cost constraint:

$$\max_{\pi} \mathbb{E}_{s \sim \mu_0}\left[V_r^{\pi}(s)\right] \quad \text{subject to} \quad V_c^{\pi} \leq \kappa, \tag{1}$$

where $\kappa \geq 0$ is a user-specified safety cost threshold (also referred to as a cost budget or limit).

*Tight cost limits* (i.e., small $\kappa$ values) present an especially demanding regime where many existing OSRL methods falter. In such safety-critical settings, such as autonomous driving, robotics, or industrial control, even minor constraint violations can be unacceptable. Common approaches based on constrained optimization or dual updates often face practical implementation challenges, such as high sensitivity to tuning, which can result in unstable training dynamics or overly conservative policies. Our goal is to develop a minimalist approach that can be wrapped around existing offline RL methods to solve OSRL problems under the challenging setting of small cost budgets.

## 3 RELATED WORK

**Online Safe RL.** Safety in online RL has been widely studied (García & Fernández, 2015; Gu et al., 2024; Wachi et al., 2024). In this setting, the agent interacts with the environment while adhering to safety constraints. A common mechanism is penalty-based control, shaping the reward as

$r' = r - \lambda \cdot c$. While simple, fixed penalties introduce a sensitive trade-off between reward and constraint satisfaction, and often require extensive tuning; e.g., RCPO (Tessler et al., 2018) and Safety Gym (Ray et al., 2019) include baselines with fixed $\lambda$ values, with main results based on adaptive Lagrangian updates. ROSARL formalizes penalty selection via the "minmax penalty" (Tasse et al., 2023). Within this penalty-based family, Sauté RL (Sootla et al., 2022) augments the state with a safety budget, and MASE (Wachi et al., 2023) casts online safe exploration as a generalized problem that combines uncertainty quantification with a reset action to prevent violations during training. Recent work also targets adaptability under changing constraints, e.g., constraint-conditioned value function approximation (Yao et al., 2023).

**Offline RL.** Offline reinforcement learning focuses on learning policies purely from fixed datasets without additional interaction with the environment (Levine et al., 2020; Figueiredo Prudencio et al., 2024). A key challenge in this setting is distributional shift between the behavior policy and the learned policy. Approaches addressing this include value estimation regularization (Fujimoto & Gu, 2021; Kostrikov et al., 2022; Kumar et al., 2019; Lyu et al., 2022; Yang et al., 2022), generative or sequential modeling (Janner et al., 2021; Wang et al., 2022), and uncertainty-aware learning (An et al., 2021; Bai et al., 2022). Some techniques utilize Q-function based action selection via filtering and reweighting, including SfBC and IDQL (Chen et al., 2023; Hansen-Estruch et al., 2023). Other methods constrain policy updates using divergence measures (Wu et al., 2020; Jaques et al., 2020; Wu et al., 2022), or leverage advances in model-based RL (Kidambi et al., 2020; Yu et al., 2020; Rigter et al., 2022) and imitation learning (Xu et al., 2022a).

**Offline Safe RL.** OSRL extends the offline RL setting to include user-defined cost constraints (Liu et al., 2024). Many OSRL methods rely on constrained optimization, often using Lagrangian relaxation (Xu et al., 2022b; Lee et al., 2022; Polosky et al., 2022), or exploiting convexity assumptions in cost and reward trade-offs (Zhang et al., 2024). However, they often require solving interdependent optimization problems and are prone to instability, particularly under strict cost limits.

Beyond constrained optimization, several recent methods propose alternative strategies. FISOR (Zheng et al., 2024) enforces safety via diffusion models to select only feasible actions and is the only method designed to handle the challenging setting of small cost budgets. TraC (Gong et al., 2025) introduces a trajectory-based classification method for safe policy learning. Latent safety modeling was also explored in (Koirala et al., 2024). LSPC employs a conditional VAE to encode conservative safety constraints into a latent space and perform reward optimization via advantage-weighted regression in that space. CAPS (Chemingui et al., 2025) switches between pre-trained policies to adapt to different test-time cost constraints. (Guo et al., 2025) propose a constraint-conditioned actor-critic (CCAC) method that explicitly models the relationship between state-action distributions and constraints, to improve generalization to unseen cost thresholds.

Diffusion-based generative models have also gained traction for OSRL. TREBI (Lin et al., 2023) employs trajectory-level diffusion sampling guided by safety classifiers, while other works adapt models such as Decision Transformer (Chen et al., 2021) and Diffuser (Janner et al., 2022) to the constrained setting (Liu et al., 2023; Lin et al., 2023). While powerful, these methods often require additional architectural components, hyperparameter tuning, or auxiliary optimization targets.

The overall goal of this paper is to address two key limitations of prior work. First, there is very little work on OSRL under small cost budgets. FISOR produces safe policies in this regime, but it achieves low reward. Second, Lagrangian-based constrained policy optimization methods can be difficult to stabilize in offline settings, often requiring extensive tuning to achieve optimal performance. The proposed constraint-aware reward relabeling approach is minimalist, can be wrapped around offline RL algorithms.

## 4 STATE-ACTION-WISE CONSTRAINTS FOR SAFETY

This section first describes an alternative formulation which enforces safety for all states and theoretically shows that its solution also solves the original problem which enforces safety constraint in expectation. Next, we provide the sketch of an iterative policy improvement approach that is most natural to solve our formulation and forms the basis for our proposed algorithm.

**Formulation for state-action-wise safety.** Solving the constrained MDP in Equation (1) often involves reformulating it as an unconstrained optimization problem via the Lagrangian method.

Although this reformulation offers a principled approach, it requires an intricate tuning of the Lagrangian multiplier, where the final performance is sensitive to it. To overcome this challenge, we consider a stronger formulation that aims to ensure constraint satisfaction for all state-action pairs a policy will encounter which is particularly beneficial when the cost limit $\kappa$ is small.

$$\max_{\pi} \; V_r^{\pi} \text{ subject to } Q_c^{\pi}(s, \pi(s)) \leq \kappa, \forall s. \tag{2}$$

where the $\max$ operator over value functions considers $V_1 \leq V_2$ if $V_1(s) \leq V_2(s), \forall s$. The above pointwise constraints ensure that we enforce safety across all states and actions that the policy will select, not simply over the expected value of the cost function as in the optimization problem from Equation (1). A solution to the optimization problem in Equation (2), if it exists, immediately yields a solution to the problem in Equation (1), but not vice versa.

The point-wise constraints formulation offers various benefits. First, to a certain extent, the point-wise constraint should be a preferred solution to many real-world applications that require safety. It requires that, no matter where we start within the system, the expected cumulative cost will be within the safety threshold. On the other hand, the problem in Equation (1) only requires that a policy is safe in the expectation with respect to the initial state and action. This might not be desirable. In the deployment of safe RL for real safety-critical applications, we are often in a one-shot setting, instead of performing repeated experiments. In such a case, only the solution for Equation (2) but not from Equation (1) guarantees safety every time during deployment.

Second, and most importantly, the point-wise constraints allow us to turn the offline constrained RL problem into an unconstrained optimization problem where it is free of tuning Lagrangian multipliers. Additionally, this unconstrained optimization naturally motivates a simple iterative algorithm that allows us to leverages powerful off-the-shelf solvers. Let us consider the following problem:

$$\max_{\pi} \; V_{r_\pi}^{\pi} \quad \text{where} \quad r_\pi(s, a) := 1_{\{Q_c^{\pi}(s,a) \leq \kappa\}} \cdot r(s, a) - 1_{\{Q_c^{\pi}(s,a) > \kappa\}} V_{\max} \tag{3}$$

with $V_{\max} = R_{\max}/(1 - \gamma)$ being the maximum possible infinite-horizon value. We show in the Theorem below that it suffices to solve the unconstrained optimization in Equation (3).

**Theorem 1.** Assume there exists a solution to Problem (2). Then a policy $\pi^*$ is an optimal solution to Problem (2) if and only if it is an optimal solution to the unconstrained optimization in (3).

**Proof.** Consider any safe policy $\pi$ according to the constraint in Problem (2). By definition we know that for each state $s$, $Q_c^{\pi}(s, \pi(s)) \leq \kappa$. This means that for any state-action pair $(s_t, a_t)$ along a trajectory $\tau$ generated by $\pi$, we have $r(s_t, a_t) = r_\pi(s_t, a_t)$. Thus, for any safe policy $\pi$ we have $V_r^{\pi} = V_{r_\pi}^{\pi}$. To complete the proof we show that any solution to Problem (3) must be safe.

Let $\tilde{\pi}^*$ be a solution to Problem (2). Let $\pi^*$ be a solution to Problem (3). We will show that $\pi^*$ must satisfy the point-wise safety, i.e., $Q_c^{\pi^*}(s, \pi^*(s)) \leq \kappa, \forall s$. Indeed, assume that $Q_c^{\pi^*}(s, \pi^*(s)) > \kappa$ for some $s$. Then, $r_{\pi^*}(s, \pi^*(s)) = -V_{\max}$. Thus, $V_{r_{\pi^*}}^{\pi^*}(s) = -V_{\max} + \mathbb{E}_{\pi^*, P}\left[\sum_{t=1}^{\infty} \gamma^t r_{\pi^*}(s_t, a_t)\right] < 0 < V_r^{\tilde{\pi}^*}(s) = V_{r_{\tilde{\pi}^*}}^{\tilde{\pi}^*}(s)$, where the last equality follows from the safety of $\tilde{\pi}^*$. This contradicts that $\pi^*$ is an optimal policy to Problem (3). Thus, $Q_c^{\pi^*}(s, \pi^*(s)) \leq \kappa, \forall s$ and hence is safe according to Problem (2). $\square$

**Sketch of an Iterative Policy Improvement Algorithm.** Motivated by the classical policy iteration method, we can design a policy iteration variant to solve the unconstrained optimization in Equation (3) as summarized below:

$$\pi_t \xrightarrow{\text{offline policy evaluation}} Q_c^{\pi_t} \xrightarrow{\text{reward relabeling}} r_{\pi_t} \xrightarrow{\text{offline policy optimization}} \pi_{t+1} \tag{4}$$

The goal of this iterative method is to improve utility and safety trade-offs of the policy incrementally from one iteration to the next. Specifically, in each iteration $t$, given the current policy $\pi_t$, we estimate the cost function $Q_c^{\pi_t}$ using an offline policy evaluation (OPE) solver. Next, we compute the reshaped reward $r_{\pi_t}$ based on $Q_c^{\pi_t}$. Finally, we perform offline policy optimization (OPO) by calling an offline RL algorithm with the reward-relabeled offline data $(s, a, r_{\pi_t}, s')$ to obtain a improved policy $\pi_{t+1}$. Building on this general principle, we describe a concrete algorithmic approach next.

## 5 CARL: CONSTRAINT-AWARE REWARD RELABELING

Our solution approach is based on designing a wrapper around any *batch-update* offline RL algorithm. In particular, batch update algorithms can be described via batch update rules, where $\mathcal{A}_{\mathrm{OPE}}$ denotes an offline policy evaluation update rule, and $\mathcal{A}_{\mathrm{OPO}}$ an offline policy optimization update rule. Each accepts a mini-batch $\mathcal{B} = \{(s_i, a_i, r_i, s_i')\}$ and performs:

$$Q' \leftarrow \mathcal{A}_{\mathrm{OPE}}(Q, \pi, \mathcal{B}), \qquad (\pi', Q') \leftarrow \mathcal{A}_{\mathrm{OPO}}(\pi, Q, \mathcal{B}),$$

allowing iterative improvement by repeatedly sampling batches from a given offline training dataset. Most state-of-the-art offline RL algorithms—e.g., TD3-BC, IQL—can be expressed in this form. The goal is to wrap these existing update functions so that, given a safe offline RL problem in the form of a training dataset $\mathcal{D}$ and a cost budget $\kappa$, the resulting policy $\pi$ maximizes reward while satisfying the safety constraint.

Below we first describe the motivation for our approach, followed by the algorithmic details.

### 5.1 ACTION FILTER MOTIVATION

To motivate our approach, consider a finite MDP with discrete states and actions. We can view applying the iteration in Equation 4 to this MDP as integrating an *action filter* into a standard unconstrained solver (e.g., policy iteration): given the current policy $\pi$, estimate its cost-to-go $Q_c^\pi(s, a)$, and remove from the MDP all actions whose cost-to-go exceeds the budget $\kappa$. Solving this reduced MDP yields a new policy $\pi'$, and the process can be repeated, always starting each iteration with the *full* action set. Across iterations, actions previously removed may be reintroduced if they become safe, and safe actions may be removed if they become unsafe.

While intuitive, this process can be unstable. The root cause is that after each policy update, the cost-to-go function can change drastically, causing the set of filtered actions to vary arbitrarily from one iteration to the next.

One way to mitigate this instability is to update the policy and cost-to-go *gradually*, so that they track each other closely throughout the optimization process. In the discrete MDP setting, this might mean updating only for a small batch of states at each iteration. Below we extend this idea to the more general setting of continuous state and action *offline* safe RL.

### 5.2 FILTERING VIA BATCH REWARD RELABELING

In continuous domains with function approximation, "removing" an individual unsafe action is ill-posed: we instead want to suppress an *entire neighborhood* of similar actions. A simple way to achieve this is via the reward relabeling approach implicit in Problem (3) where we replace the reward for any state–action pair predicted to be unsafe with a maximally negative constant. Function approximation then naturally generalizes this penalty to nearby actions, discouraging them without needing to identify and prune them explicitly.

We use a simple approach to integrate reward relabeling and batch-update offline RL. Specifically, given the current Q-cost function $Q_c^\pi$ under the current policy $\pi$, we define the *constraint-aware reward relabeling rule* for a transition $(s, a, r, s')$ as:

$$\mathrm{CARL}(s, a, r, s') = \begin{cases} (s, a, r, s'), & \text{if } Q_c^\pi(s, a) \leq \kappa, \\ (s, a, -V_{\max}, s') & \text{otherwise,} \end{cases} \tag{5}$$

Performing batch updates where rewards in each batch are relabeled naturally avoids actions currently considered unsafe and, by generalization, actions similar to them.

The generic batch reward relabeling loop alternates between the following two steps:

1. **Cost evaluation:** $M$ iterations of OPE updates on $(s, a, c, s')$ to refine $Q_c^\pi$.

2. **Policy optimization:** $K$ iterations of OPO updates on $\mathrm{CARL}(s, a, r, s')$ given by 5.

---

**Algorithm 1** Constraint-aware Reward Relabeling (CARL)

---

**Require:** Offline dataset $\mathcal{D}$, budget $\kappa$, backbone updates $\mathcal{A}_{\text{OPE}}, \mathcal{A}_{\text{OPO}}$, batch size $m$
1: Init cost critic $Q_c$, policy $\pi$, and reward critic $Q_r$
2: **while** not converged **do**
3:     Sample mini-batch $\{(s_i, a_i, r_i, c_i, s_i')\}_{i=1}^m \subset \mathcal{D}$
4:     $Q_c \leftarrow \mathcal{A}_{\text{OPE}}\big(Q_c, \pi, \{(s_i, a_i, c_i, s_i')\}_{i=1}^m\big)$
5:     $(\pi, Q_r) \leftarrow \mathcal{A}_{\text{OPO}}\big(\pi, Q_r, \{\text{CARL}(s_i, a_i, r_i, s_i')\}_{i=1}^m\big)$
6: **end while**
7: **return** $\pi$

---

Intuitively, large $M$ and $K$ let $Q_c$ and $\pi$ change substantially between phases. Thus, as described above for discrete MDPs, this can cause severe oscillations: the algorithm alternates between unsafe, high-reward policies and overly conservative safe policies, never exploring the many safe policies with high reward. Figure 1 illustrates an implementation based on TD3BC where we observe this oscillation for a standard benchmark problem AntRun.

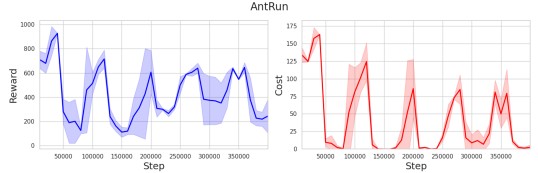

Figure 1: Oscillatory Performance of `AntRun`: reward (left) and cost (right) across training steps with a cost limit of 40.

To address this instability, a natural stabilizing choice is to keep $M$ and $K$ *small* so that $Q_c$ and $\pi$ track each other closely. The most extreme case, $M = K = 1$, suggests a simple wrapper with no additional tunable hyperparameters (utilizing dataset-derived penalties) beyond the base OPE and OPO algorithms. Each iteration involves sampling a mini-batch $B$ followed by:

$$\text{Update } Q_c^\pi \text{ with } B \quad \longrightarrow \quad \text{Relabel } B \text{ via Equation 5} \quad \longrightarrow \quad \text{Update } \pi \text{ with relabeled } B.$$

The pseudo-code for this incremental variant called is shown in Algorithm 1. Despite the minimalism, we find that $M = K = 1$ consistently results in state-of-the-art performance. Further, while one can treat $K$ and $M$ as hyperparameters, we have not found values that consistently outperform CARL across benchmarks. Because the CARL algorithm modifies only the reward before passing data to the backbone, it inherits the underlying offline RL algorithms' sample efficiency and out-of-distribution handling.

While setting $K = M = 1$ greatly reduces drift compared to using large values of $K$ and $M$, theoretical convergence guarantees are unclear. Formally analyzing whether $K = M = 1$ converges—possibly under assumptions on the MDP class, dataset coverage, or backbone stability—is an open problem. Empirically, we find the method to be stable across all tested benchmarks and initializations, and achieving significantly better performance than state-of-the-art offline safe RL methods.

**Summary of CARL's advantages.** Our proposed CARL approach has the following advantages.

- It can be wrapped around existing offline RL algorithms to effectively solve offline safe RL problems as we demonstrate in our experiments.
- It is a minimalist approach that doesn't introduce any additional tunable hyperparameters.

## 6 EXPERIMENTS AND RESULTS

### 6.1 EXPERIMENTAL SETUP

**Benchmarks.** We evaluate CARL across a variety of offline safe reinforcement learning tasks using the DSRL benchmark suite (Liu et al., 2024), which offers standardized datasets with diverse safety constraints. Our evaluation mainly spans both the Bullet-based environments (Car-Run, Drone-Circle etc.), as well as broader generalization through additional tasks drawn from Safety-Gymnasium (Ray et al., 2019; Ji et al., 2024). The combined benchmark suite includes diverse tasks spanning a range of difficulty levels and safety complexities.

**Evaluation Protocol.** Consistent with DSRL standards, we report two core metrics: normalized cumulative reward and normalized cumulative cost. For a task $T$, if $r_{\max}(T)$ and $r_{\min}(T)$ represent the maximum and minimum observed cumulative rewards, the normalized reward for a policy $\pi$ is computed as:

$$R_{\text{norm}} = \frac{R_\pi - r_{\min}(T)}{r_{\max}(T) - r_{\min}(T)}, \qquad C_{\text{norm}} = \frac{C_\pi}{\kappa}.$$

where $C_\pi$ is the policy's cumulative cost and $\kappa$ is the cost threshold. A policy is considered safe if $C_{\text{norm}} \leq 1$. This formulation differs from alternative evaluations in CCAC (Guo et al., 2025) that normalize rewards only using trajectories satisfying the cost budget $\kappa$. Instead, we follow the standard DSRL evaluation protocol, which uses the full reward range per task. Our main results use stringent thresholds of $\kappa = 5$ and $\kappa = 10$ to test performance in highly constrained scenarios. Each policy is tested over 20 episodes and averaged over three random seeds.

**Baselines.** We compare CARL against a range of state-of-the-art OSLR methods. `BC-Safe` is a behavior cloning baseline that trains on trajectories satisfying a predefined cost threshold. `CPQ` (Xu et al., 2022b) penalizes out-of-distribution actions and updates Q-values using only safe transitions. `COptiDICE` (Lee et al., 2022) addresses safety via stationary distribution correction. `CDT` (Liu et al., 2023) uses a decision transformer to learn policies conditioned on reward and cost enabling test-time constraint adaptation. `CAPS` (Chemingui et al., 2025) also supports test-time constraint handling by switching among multiple pre-trained policies. `FISOR` (Zheng et al., 2024) is a diffusion-based approach that directly optimizes for feasibility to produce zero-violation policies. Finally, we include `CCAC` (Guo et al., 2025), a recent method that uses a constraint-conditioned actor–critic with generative modeling and OOD detection for adaptive, safe policy learning under varying constraints. We further evaluated Lagrangian variants of offline RL algorithms in Table 5.

Our results use CARL as a wrapper around TD3-BC, with fitted-Q evaluation (FQE) employed for off-policy evaluation (Le et al., 2019). Implementation details are available in the Appendix.

## 6.2 Results and Discussion

**CARL vs. Baselines.** Table 1 compares CARL with a suite of offline safe RL baselines across 19 DSRL tasks. Performance is reported in terms of normalized reward (higher is better) and normalized cost (lower is better, with values $\leq 1$ indicating constraint satisfaction) under low budgets of 5 or 10. For the main results, we set the penalty using $R_{\max} = \max_{(s,a,r,\cdot)} r$ from the offline data instead of $V_{\max}$; an ablation with the larger penalty $V_{\max}$ is included in Table 5 in the appendix. Remarkably, CARL is the *only* method that satisfies the cost constraint across *all* Bullet tasks. It is also safe on 8 out of 11 in the more challenging SafetyGym tasks. While other methods such as CAPS, FISOR, or CCAC manage to remain safe on a few tasks, none of them achieve the same level of consistency for safety. Importantly, CARL's safety does not come at the expense of reward performance. CARL consistently ranks as the best or second-best safe method in terms of reward. While other baselines occasionally achieve higher returns, the top-performing method varies across tasks and fails to consistently satisfy the cost constraint, unlike CARL which maintains stricter safety in most of the tasks.

These results demonstrate that CARL strikes a strong balance between reward maximization and constraint satisfaction, without requiring special reward shaping, risk-sensitive training objectives, or task-specific tuning. Its consistent safety and competitive performance across tasks and cost budgets highlight its robustness and suitability for safety-critical settings.

**Backbone Offline RL Algorithm.** CARL can be wrapped around any off-the-shelf offline RL algorithm without modifying the method's loss, targets, or regularizers. Our main results use TD3-BC (Fujimoto & Gu, 2021) as the backbone. To test generality, we also evaluate CARL with IQL (Kostrikov et al., 2022), which differs significantly in design. TD3-BC is an actor–critic method that queries the current policy to generate target actions and applies behavior cloning regularization on the actor. In contrast, IQL estimates Q-values purely from dataset actions and optimizes the policy separately via advantage-weighted regression, without querying the policy during value learning.

As shown in Table 2, CARL maintains safety and achieves comparable rewards under both backbones. This indicates that our relabeling in Equation 5 is agnostic to the underlying backbone, confirming that CARL generalizes effectively across offline RL algorithms.

Table 1: CARL main results for normalized rewards and costs. The normalized cost threshold is 1 ($\kappa = 5$ or $10$). The ↑ symbol denotes that higher rewards are better. The ↓ symbol denotes that lower costs (up to threshold 1) are better. Each cell shows reward or cost with standard deviation. **Bold**: safe agents ($C_{\text{norm}} \leq 1$). Gray: unsafe agents. **Blue bold**: best safe result ($C_{\text{norm}} \leq 1$).

| Tasks | | BC Safe | CPQ | COptiDICE | CDT | CAPS | CCAC | FISOR | CARL |
|---|---|---|---|---|---|---|---|---|---|
| **Bullet Gym Tasks:** $\kappa = 5$ | | | | | | | | | |
| BallRun | Reward ↑ | 0.16±0.11 | 0.09±0.26 | 0.53±0.10 | 0.27±0.09 | **0.07±0.05** | **0.31±0.01** | 0.09±0.07 | **0.28±0.02** |
| | Cost ↓ | 4.50±3.38 | 2.20±2.88 | 10.83±1.68 | 2.57±3.23 | **0.00±0.00** | **0.00±0.00** | 1.28±1.70 | **0.00±0.00** |
| CarRun | Reward ↑ | **0.92±0.01** | **0.93±0.01** | 0.91±0.04 | **0.99±0.00** | **0.97±0.00** | 1.82±0.84 | **0.74±0.01** | **0.97±0.00** |
| | Cost ↓ | **0.26±0.23** | **0.20±0.18** | 0.00±0.00 | 0.90±0.34 | **0.11±0.17** | 24.57±20.94 | **0.00±0.00** | **0.02±0.03** |
| DroneRun | Reward ↑ | 0.41±0.23 | 0.29±0.11 | 0.68±0.00 | **0.58±0.00** | 0.41±0.06 | 0.50±0.08 | 0.31±0.04 | **0.36±0.12** |
| | Cost ↓ | 1.62±1.73 | 2.35±4.08 | 15.02±0.10 | **0.07±0.07** | 5.70±3.08 | 16.29±9.35 | 2.52±1.10 | **0.30±0.52** |
| AntRun | Reward ↑ | 0.56±0.02 | **0.03±0.05** | 0.61±0.01 | 0.70±0.03 | 0.53±0.13 | **0.03±0.10** | **0.43±0.02** | **0.36±0.09** |
| | Cost ↓ | 1.15±0.47 | **0.05±0.08** | 3.26±1.39 | 1.66±0.24 | 2.03±2.17 | **0.00±0.00** | **0.27±0.15** | 0.60±0.41 |
| BallCircle | Reward ↑ | 0.45±0.03 | 0.56±0.10 | 0.71±0.01 | 0.61±0.17 | **0.33±0.02** | 0.47±0.38 | **0.32±0.05** | **0.69±0.03** |
| | Cost ↓ | 1.21±0.23 | 1.11±1.92 | 9.41±0.69 | 2.03±1.39 | **0.01±0.02** | 10.19±17.65 | **0.00±0.00** | **0.33±0.23** |
| CarCircle | Reward ↑ | 0.31±0.06 | **0.71±0.02** | 0.49±0.01 | 0.71±0.01 | **0.40±0.03** | 0.70±0.01 | **0.37±0.02** | **0.69±0.01** |
| | Cost ↓ | 1.57±0.07 | **0.00±0.00** | 10.82±1.28 | 1.58±0.57 | **0.03±0.03** | 1.61±0.51 | **0.00±0.00** | **0.00±0.00** |
| DroneCircle | Reward ↑ | 0.50±0.01 | -0.21±0.04 | 0.26±0.02 | 0.55±0.01 | **0.36±0.02** | 0.43±0.04 | **0.48±0.01** | **0.53±0.02** |
| | Cost ↓ | 1.22±0.31 | 0.70±0.41 | 3.68±0.51 | 1.14±0.06 | **0.00±0.00** | 0.06±0.07 | **0.17±0.15** | **0.00±0.00** |
| AntCircle | Reward ↑ | 0.40±0.02 | 0.00±0.00 | 0.18±0.00 | 0.45±0.05 | **0.33±0.05** | 0.49±0.07 | **0.24±0.02** | **0.60±0.01** |
| | Cost ↓ | 4.72±0.87 | 0.00±0.00 | 17.40±0.36 | 6.59±1.42 | **0.00±0.00** | 0.02±0.03 | **0.04±0.08** | **0.02±0.03** |
| **Safety Gym Tasks:** $\kappa = 10$ | | | | | | | | | |
| CarCircle1 | Reward ↑ | 0.27±0.09 | -0.03±0.07 | 0.68±0.05 | 0.53±0.07 | 0.42±0.04 | 0.32±0.10 | 0.69±0.03 | 0.46±0.03 |
| | Cost ↓ | 3.82±1.01 | 10.17±7.26 | 20.70±1.15 | 8.58±2.04 | 2.79±1.04 | 21.96±4.14 | 10.52±0.10 | 4.15±0.83 |
| CarCircle2 | Reward ↑ | 0.37±0.06 | 0.46±0.05 | 0.75±0.02 | 0.61±0.04 | 0.38±0.14 | 0.57±0.01 | 0.63±0.02 | 0.45±0.05 |
| | Cost ↓ | 6.90±2.62 | 2.41±3.95 | 26.40±2.30 | 13.12±2.59 | 4.15±5.07 | 16.85±4.07 | 12.78±1.97 | 1.57±1.38 |
| CarGoal1 | Reward ↑ | **0.20±0.09** | 0.67±0.07 | 0.40±0.03 | 0.62±0.04 | 0.27±0.00 | 0.82±0.06 | 0.42±0.07 | **0.26±0.04** |
| | Cost ↓ | **0.40±0.12** | 4.41±0.46 | 2.50±0.36 | 3.12±0.69 | 1.15±0.10 | 5.27±0.92 | 1.70±0.58 | **0.92±0.55** |
| CarGoal2 | Reward ↑ | 0.14±0.02 | 0.53±0.14 | 0.21±0.10 | 0.42±0.06 | 0.11±0.06 | 0.91±0.02 | **0.05±0.01** | 0.13±0.03 |
| | Cost ↓ | 1.45±0.60 | 15.09±1.91 | 3.01±1.45 | 5.34±1.25 | 2.05±2.23 | 15.80±1.04 | **0.50±0.71** | 1.77±0.51 |
| PointCircle1 | Reward ↑ | **0.30±0.11** | 0.46±0.04 | 0.87±0.01 | **0.54±0.01** | 0.31±0.05 | 0.57±0.08 | 0.43±0.05 | **0.52±0.01** |
| | Cost ↓ | **0.26±0.24** | 0.73±1.20 | 19.16±0.38 | **0.54±0.61** | 0.94±1.53 | 6.65±3.51 | 14.93±4.24 | **0.04±0.07** |
| PointCircle2 | Reward ↑ | 0.42±0.07 | 0.44±0.05 | 0.85±0.01 | 0.59±0.02 | **0.44±0.06** | 0.03±0.75 | 0.76±0.05 | **0.55±0.01** |
| | Cost ↓ | 2.10±0.37 | 1.20±1.28 | 29.36±1.23 | 2.05±0.93 | **0.10±0.09** | 3.99±4.30 | 18.02±4.17 | **0.91±1.46** |
| PointGoal1 | Reward ↑ | **0.22±0.02** | 0.55±0.11 | 0.49±0.02 | 0.63±0.08 | 0.30±0.09 | 0.74±0.03 | 0.64±0.03 | **0.06±0.06** |
| | Cost ↓ | **0.91±0.42** | 3.33±1.68 | 5.27±0.91 | 2.97±0.82 | 1.45±0.74 | 3.90±0.50 | 5.38±1.05 | **0.09±0.11** |
| PointGoal2 | Reward ↑ | 0.18±0.01 | 0.25±0.13 | 0.41±0.06 | 0.46±0.08 | 0.19±0.04 | 0.78±0.05 | 0.31±0.07 | **0.13±0.05** |
| | Cost ↓ | 1.81±0.72 | 5.34±1.33 | 5.68±0.48 | 5.57±0.81 | 1.02±0.22 | 15.51±5.75 | 1.67±1.32 | **0.81±0.31** |
| AntVelo | Reward ↑ | **0.92±0.03** | -1.01±0.00 | 1.00±0.01 | **0.97±0.02** | **0.88±0.05** | -1.01±0.00 | **0.89±0.01** | **0.99±0.01** |
| | Cost ↓ | **0.33±0.19** | 0.00±0.00 | 11.82±5.56 | **0.59±0.02** | **0.21±0.07** | 0.00±0.00 | **0.00±0.00** | **0.43±0.11** |
| HalfCheetahVelo | Reward ↑ | **0.86±0.01** | 0.67±0.05 | **0.63±0.02** | **0.98±0.00** | **0.86±0.01** | **0.91±0.04** | **0.89±0.01** | **0.96±0.01** |
| | Cost ↓ | **0.30±0.13** | 8.56±6.19 | **0.00±0.00** | **0.65±0.34** | **0.27±0.06** | **0.97±0.12** | **0.00±0.00** | **0.14±0.09** |
| SwimmerVelo | Reward ↑ | **0.46±0.07** | 0.20±0.20 | 0.59±0.07 | 0.66±0.01 | 0.45±0.06 | **0.06±0.25** | -0.02±0.05 | 0.21±0.19 |
| | Cost ↓ | **0.80±0.19** | 2.82±4.19 | 16.80±7.39 | 1.26±0.50 | 2.17±2.91 | **0.91±1.57** | 0.23±0.20 | **0.00±0.00** |

Table 2: Comparison of TD3BC and IQL across Bullet Safety Gym and Velocity tasks. Each cell shows reward (↑) or cost (↓) with standard deviation. **Bold**: safe agents ($C_{norm} \leq 1$).

| Method | | CarRun | DroneRun | CarCircle | DroneCircle | AntVelocity | HalfCheetahVelo |
|---|---|---|---|---|---|---|---|
| TD3BC | Reward ↑ | **0.97±0.00** | **0.36±0.12** | **0.69±0.01** | **0.53±0.02** | **0.99±0.01** | **0.96±0.01** |
| | Cost ↓ | **0.02±0.03** | **0.30±0.52** | **0.00±0.00** | **0.00±0.00** | **0.43±0.11** | **0.14±0.09** |
| IQL | Reward ↑ | **0.96±0.01** | **0.46±0.03** | **0.68±0.01** | **0.35±0.02** | **0.97±0.01** | **0.92±0.02** |
| | Cost ↓ | **0.12±0.14** | **0.71±1.06** | **0.08±0.10** | **0.00±0.00** | **0.38±0.09** | **0.26±0.40** |

**Evaluation Across Varying Cost Limits.** While our main results are for the strict cost thresholds ($\kappa$ equals 5 or 10) representative of safety-critical settings, we also evaluate CARL under more relaxed safety constraints: cost limits of 20 and 40 for Bullet tasks, and 40 and 80 for Safety Gym tasks, following the setup introduced in DSRL. Based on the results shown in Table 1, the safest baseline methods after CARL are FISOR, CAPS, CPQ, and CCAC. However, FISOR is trained solely to minimize cost and does not adapt to different cost limits. CPQ, while safe in some tasks, achieves

very low rewards in two of the safe results. Therefore, for the varying cost limit analysis, we focus our comparison on CAPS and CCAC, both of which show better balance for safety and reward performance.

Figures 2 show that CARL improves rewards as the cost budget increases, while keeping normalized costs within the safety threshold ($\leq 1$). On the challenging **CarCircle2** task, all methods are unsafe at budget 10. When the budget rises to 40 or 80, CARL attains both safety and higher rewards, whereas CAPS and CCAC remain unsafe (Table 6, Appendix). This demonstrates CARL's ability to exploit larger budgets where other methods fail.

**Training Only on Unsafe Trajectories.** We perform an ablation where we train CARL using either the full dataset, or only unsafe trajectories (those with cumulative cost exceeding the threshold ($\kappa = 5$ or 10)).

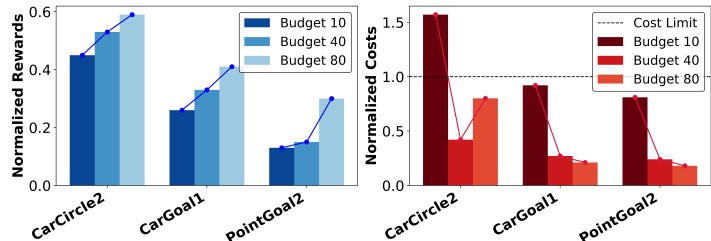

Figures 3 shows results for training with only unsafe trajectories (blue). CARL generates 60 trajectories per task (20 from each of the three seeds), shown in red. Despite the absence of safe training examples, these trajectories remain within the cost threshold (dashed line) while achieving strong rewards. In BALLCIRCLE, CARL attains rewards around 600–650, enforcing safety with little loss in rewards. In ANTVE-LOCITY, it reaches near-optimal rewards of ∼3000 while staying safe, combining strict constraint satisfaction with top-level performance. In ANTCIRCLE, CARL shifts trajectories into the safe zone while still reaching rewards up to 300+, comparable to the best unsafe cases near the boundary, showing that meaningful reward performance can be retained under strict safety. Overall, these results highlight CARL's ability to recover safe and competitive behavior from purely unsafe data through reward relabeling: transforming unsafe dataset trajectories into safe ones and shifting behavior into the feasible region without substantial loss in rewards.

Figure 2: Results for CARL with varying cost budget limits on three tasks: rewards vs. cost budget (left) and costs vs. cost budget (right).

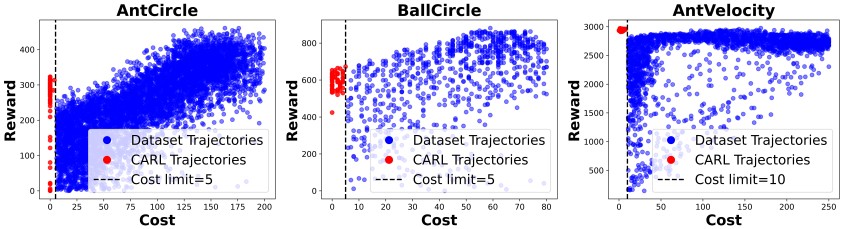

Figure 3: Results for CARL-generated trajectories (red) when trained on unsafe data, compared against unsafe dataset trajectories (blue) on three representative tasks.

We also evaluate a naive hard-filtering variant (Appendix Table 8), which removes unsafe transitions entirely, and find that it fails on nearly all tasks, underscoring the importance of CARL's reward relabeling rather than simple data exclusion.

## 7 SUMMARY

This paper introduces an embarrassingly simple framework for solving offline safe reinforcement learning (OSRL). It reformulates the OSRL problem into an unconstrained optimization problem that requires no tuning of Lagrangian multipliers when compared to the vast majority of the previous methods. This framework allows us to naturally leverage powerful off-the-shelf offline RL algorithms and advances directly to effectively solve OSRL problems via constraint-aware reward relabeling. Experimental results on the DSRL benchmark demonstrate the remarkably strong performance of the proposed framework compared to state-of-the-art OSRL methods.

**Reproducibility Statement.** We include full implementation details in Appendix C, including hyperparameters and training settings. All datasets used are publicly available. An anonymous link to the source code is provided in the abstract to support reproducibility of our experiments.

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

## A EXTENDED RELATED WORK DISCUSSION

A recurring theme across safe reinforcement learning is the idea of discouraging unsafe actions through penalties, though the way those penalties are implemented differs across methods. In the online setting, MASE (Wachi et al., 2023) integrates uncertainty quantification with an emergency stop: if no action can be certified safe, the episode is halted and a large penalty is applied. Sauté RL Sootla et al. (2022) also relies on penalties but does so through a state augmentation mechanism: each state is paired with a remaining safety budget, and once the budget runs out, the trajectory transitions into an absorbing unsafe state with a per task tuned large penalty. This budget-based reformulation means that penalties are not just added externally but become part of the dynamics, shaping exploration as the agent trains. In both approaches, penalties are tightly coupled with the exploration process, ensuring that unsafe behavior is discouraged while the agent is still gathering data.

CARL operates under a different contract. In the offline setting, trajectories are fixed in advance: there is no opportunity to terminate episodes at will, and unsafe trajectories cannot simply be avoided during training. The evaluation of a policy must account for the costs accumulated across complete episodes, regardless of whether individual transitions are safe or not. This creates a distinctive challenge: unsafe transitions are often plentiful in available datasets, and discarding them reduces coverage, while treating them naively may compromise deployment safety. CARL addresses this challenge by transforming unsafe transitions into useful training signal. Rather than filtering or generating alternative trajectories, it directly relabels rewards with large penalties whenever the cost critic predicts that a safety budget would be violated. This allows learning to leverage the entire dataset, including unsafe portions, while still driving the learned policy toward safe behavior.

Placed in the broader landscape of offline safe RL, this perspective contrasts with approaches that construct explicit feasibility regions Zheng et al. (2024), rely on constraint-conditioned generative models (Guo et al., 2025), or employ diffusion-based feasibility shaping (Yao et al., 2024). Unlike CPQ (Xu et al., 2022b), which masks the policy and zeroes out future reward value estimates for unsafe actions, CARL employs active reward penalization to provide a continuous, negative gradient signal for such unsafe actions. Those methods introduce additional modeling complexity to estimate which actions are permissible. CARL, in comparison, emphasizes a minimalist path: penalties are imposed through direct reward relabeling, and the resulting relabeled dataset can be handled by any strong offline RL backbone. Empirically, this strategy proves effective even in challenging regimes such as small cost budgets or datasets composed entirely of unsafe trajectories. In such settings, the ability to repurpose unsafe data as penalized signal is particularly advantageous, as it avoids both the brittleness of feasibility approximations and the conservatism of discarding large parts of the dataset.

From this viewpoint, the contribution of CARL is not a departure from the penalty-based philosophy shared across safe RL, but rather a demonstration that in the offline regime, the most straightforward instantiation of that philosophy, reward relabeling informed by cost estimates,can be both robust and broadly applicable. This observation highlights that in certain cases, simplicity in how penalties are enforced may offer a more reliable path to safety than more sophisticated mechanisms.

## B ADDITIONAL RESULTS

### B.1 ROBUSTNESS TO OPE ESTIMATION ERRORS

Accurate off-policy evaluation (OPE) is a challenge for all offline safe RL methods that rely on value estimation. However, the mechanism by which estimation errors affect the policy differs significantly between approaches. Lagrangian methods update dual variables based on the magnitude of the constraint violation; consequently, errors in the cost value function $Q_c$ can cause the Lagrange multiplier to grow arbitrarily large, destabilizing the policy.

In contrast, CARL employs a binary penalty mechanism: a fixed penalty is applied only if the cost limit is violated. This structure clamps the error signal, preventing estimation inaccuracies from scaling the penalty magnitude. To empirically verify this robustness, we conducted an ablation study assessing the method's sensitivity to OPE miscalibration.

Table 3: CARL performance robustness under synthetic OPE noise. The normalized cost threshold is 1. $\uparrow$ indicates higher is better (reward); $\downarrow$ indicates lower is better (cost, with values $\leq 1$ considered safe). Each cell shows the mean with standard deviation. **Bold**: safe agents ($C_{norm} \leq 1$). Gray: unsafe agents.

| | Noise level | | | | | | | |
| | 0.1 | | 0.2 | | 0.3 | | 0.5 | |
| Task | R $\uparrow$ | C $\downarrow$ | R $\uparrow$ | C $\downarrow$ | R $\uparrow$ | C $\downarrow$ | R $\uparrow$ | C $\downarrow$ |
|---|---|---|---|---|---|---|---|---|
| BallCircle | **0.73**$_{\pm 0.01}$ | **0.35**$_{\pm 0.24}$ | **0.75**$_{\pm 0.01}$ | **0.11**$_{\pm 0.05}$ | 0.49$_{\pm 0.33}$ | 2.85$_{\pm 4.61}$ | 0.87$_{\pm 0.02}$ | 10.56$_{\pm 0.81}$ |
| CarCircle | **0.70**$_{\pm 0.01}$ | **0.02**$_{\pm 0.03}$ | **0.69**$_{\pm 0.04}$ | **0.00**$_{\pm 0.00}$ | **0.71**$_{\pm 0.03}$ | **0.00**$_{\pm 0.00}$ | 0.90$_{\pm 0.01}$ | 16.73$_{\pm 0.44}$ |
| DroneCircle | **0.52**$_{\pm 0.03}$ | **0.00**$_{\pm 0.00}$ | **0.55**$_{\pm 0.01}$ | **0.00**$_{\pm 0.00}$ | -0.15$_{\pm 0.12}$ | 0.48$_{\pm 0.44}$ | 0.50$_{\pm 0.53}$ | 13.40$_{\pm 7.13}$ |
| AntCircle | **0.60**$_{\pm 0.02}$ | **0.09**$_{\pm 0.10}$ | **0.47**$_{\pm 0.05}$ | **0.00**$_{\pm 0.00}$ | **0.33**$_{\pm 0.19}$ | **0.71**$_{\pm 0.27}$ | 0.70$_{\pm 0.08}$ | 20.62$_{\pm 3.34}$ |

**Noise Injection Methodology.** During training, we randomly inverted the decision of the cost critic (flipping "safe" to "unsafe" and vice versa) with a probability $p \in \{10\%, 20\%, 30\%, 50\%\}$ to inject synthetic noise into the relabeling process.

As detailed in Table 3, our results demonstrate that the policy remains safe under moderate levels of noise. Using a normalized cost $\leq 1$ as the safety criterion, all evaluated tasks remained safe at noise levels of 10% and 20%, despite mild degradation in rewards. This indicates that reward performance deteriorates before safety is compromised, which is the intended behavior of our safety mechanism.

At 30% noise, we observed the onset of instability: BallCircle became unsafe (normalized cost 2.85) and DroneCircle suffered negative rewards ($-0.15$), while CarCircle and AntCircle remained safe. Finally, at 50% noise, all tasks incurred large safety violations (normalized costs 10–20). This regime corresponds to noise levels where the input signal is largely uninformative, resulting in behavior consistent with random inputs.

These results demonstrate that CARL does not rely on precise, pointwise accuracy of the cost estimator ($Q_c$). Instead, the method is functionally robust to significant signal degradation. Even when nearly one-third of the safety labels are noisy, the aggregate topological signal provided by the critic remains strong enough to guide the policy into the feasible region. This resilience validates the design choice of the iterative update cycle ($M = K = 1$); by co-evolving the policy and critic incrementally, transient estimation errors are mitigated over the course of training rather than accumulating catastrophically.

Furthermore, the collapse of safety at the 50% noise level serves as a critical control. It confirms that the safety observed in standard runs is causally driven by the $Q_c$ signal, and not by external factors or environment design, demonstrating that the critic is indeed providing the necessary directional guidance.

**Control Experiment: Random Penalties.** To further isolate the contribution of the cost critic, we conducted a second control experiment where we removed the $Q_c$ signal entirely. Instead of relabeling based on estimated cost, we applied penalties to a random subset of transitions (ranging from 70% to 90% of the dataset) to simulate the distribution of penalties *without* the topological information provided by the critic.

The agent failed to achieve safety across the tasks under this random penalty regime, as shown in Table 4. This failure is significant as it directly refutes the potential critique that CARL achieves safety merely through over-conservatism or indiscriminate data suppression. The results demonstrate that safety in these environments cannot be achieved simply by avoiding a large percentage of the state–action space; rather, it requires avoiding *specific*, structurally unsafe transitions.

This corroborates our hypothesis that, despite the inherent challenges of OPE, the cost critic is successfully identifying the correct manifold of unsafe state–action pairs. The penalties in CARL are not merely acting as regularization; they are effectively shaping the decision boundary. Random penalties disrupt this boundary, leading to constraint violations, whereas $Q_c$-guided penalties, even with the noise demonstrated in the previous experiment, successfully preserve it.

Table 4: Control experiment results using Random Penalties. The normalized cost threshold is 1. ↑ indicates higher is better (reward); ↓ indicates lower is better (cost, with values $\leq 1$ considered safe). Each cell shows the mean with standard deviation. **Bold**: safe agents ($C_{norm} \leq 1$). Gray: unsafe agents.

| | \multicolumn{4}{c}{**Relabeling ratio**} |
| Task | \multicolumn{2}{c}{**0.7**} | \multicolumn{2}{c}{**0.9**} |
| | R ↑ | C ↓ | R ↑ | C ↓ |
| --- | --- | --- | --- | --- |
| BallCircle | $0.54_{\pm 0.36}$ | $6.86_{\pm 1.06}$ | $0.01_{\pm 0.01}$ | $11.84_{\pm 20.50}$ |
| CarCircle | $0.84_{\pm 0.03}$ | $15.09_{\pm 0.35}$ | $0.76_{\pm 0.03}$ | $13.44_{\pm 1.33}$ |
| DroneCircle | $0.44_{\pm 0.52}$ | $8.68_{\pm 6.11}$ | $0.49_{\pm 0.53}$ | $9.49_{\pm 6.83}$ |
| AntCircle | $0.64_{\pm 0.02}$ | $18.83_{\pm 2.54}$ | $0.60_{\pm 0.07}$ | $17.47_{\pm 1.50}$ |

### B.2 PENALTY MAGNITUDE ABLATION: $R_{\max}$ VS. $V_{\max}$

Table 5 compares CARL trained with two penalty magnitudes: $R_{\max}$, the maximum single-step reward in the dataset, and $V_{\max} = R_{\max}/(1 - \gamma)$, the maximum possible discounted return. Using $R_{\max}$ produces a smaller penalty and thus a less aggressive correction, while $V_{\max}$ enforces harsher penalties. The results confirm this intuition: CARL with $V_{\max}$ remains safe in most tasks but often at the cost of lower rewards compared to the $R_{\max}$ variant. For instance, rewards drop from 0.69 $\rightarrow$ 0.50 in CARCIRCLE and 0.60 $\rightarrow$ 0.40 in ANTCIRCLE, while in DRONECIRCLE (0.53 $\rightarrow$ 0.02) and ANTVELOCITY (0.99 $\rightarrow$ 0.38) the decline is particularly severe. Nevertheless, even under the stricter $V_{\max}$ penalty, CARL outperforms other baselines: it remains safe in far more tasks, and when safe, its performance is frequently the best or second-best (e.g., BALLCIRCLE, POINTGOAL2, SWIMMERVELOCITY). Overall, $R_{\max}$ provides the best trade-off between safety and reward, and has the added advantage of not depending on the discount factor $\gamma$, thereby avoiding sensitivity to this extra hyperparameter.

To provide a broader comparison of stability, we evaluated Lagrangian extensions of offline RL methods, BEAR-Lag (Kumar et al., 2019) and BCQ-Lag (Fujimoto et al., 2019), alongside the reported baselines. As detailed in Table [REFERENCE NEW TABLE], these methods frequently failed to satisfy cost constraints: BEAR-Lag and BCQ-Lag were safe on only 2 and 1 of the 15 tasks evaluated, respectively. In comparison, CARL consistently yields safe policies. These results reinforce the observation that while Lagrangian methods possess theoretical guarantees, they are difficult to stabilize in offline scenarios, whereas CARL consistently achieves constraint satisfaction across diverse environments.

### B.3 EXTRA COST LIMITS

While our main results emphasize strict cost thresholds ($\kappa = 5, 10$), we also investigate performance under more relaxed safety budgets: cost limits of 20 and 40 for Bullet tasks, and 40 and 80 for Safety Gym tasks, following the DSRL setup. This analysis addresses a natural concern: since CARL penalizes unsafe transitions harshly, one might expect it to perform well only under tight constraints, but to underutilize more generous budgets. Table 6 shows that this is not the case. CARL continues to adapt effectively to larger budgets, often outperforming more the sophisticated methods of CAPS and CCAC.

We evaluate across 16 tasks in total, 8 from Bullet and 8 from Safety Gym, each under two cost limits, yielding 32 evaluations. Among these, CARL is the best performer in 16 cases, CAPS in 13, and CCAC in 4. CARL is unsafe only twice (both in BALLCIRCLE), whereas CAPS is unsafe in 7 cases and CCAC fails in 20.

In the Bullet tasks with larger budgets ($\kappa = 20, 40$), CARL continues to adapt effectively despite its simple penalty mechanism. On BALLRUN, CARL consistently produces safe policies, whereas CAPS and CCAC violate the cost threshold. In ANTCIRCLE, all methods remain safe, but CARL achieves the highest rewards across both budgets. The only exception is BALLCIRCLE, where CARL attains high rewards but fails to satisfy the safety constraint. Importantly, across the remaining 15 out of 16 tasks, CARL maintains safety while matching or exceeding baseline rewards. This

Table 5: CARL results with $V_{max}$ vs. $R_{max}$ penalties, reported with normalized rewards and costs. The normalized cost threshold is 1 ($\kappa = 5$ or 10). ↑ indicates higher is better (reward); ↓ indicates lower is better (cost, with values $\leq 1$ considered safe). Each cell shows the mean with standard deviation. **Bold**: safe agents ($C_{norm} \leq 1$). Gray: unsafe agents. **Blue bold**: best safe performance.

| Tasks | | BCQ-Lag | BEAR-Lag | CPQ | CDT | CAPS | CCAC | FISOR | CARL | CARL $V_{max}$ |
|---|---|---|---|---|---|---|---|---|---|---|
| **Bullet Gym Tasks:** $\kappa = 5$ | | | | | | | | | | |
| BallRun | Reward ↑ | 0.67±0.24 | 0.53±0.47 | 0.09±0.26 | 0.27±0.09 | **0.07±0.05** | **0.31±0.01** | 0.09±0.07 | **0.28±0.02** | 0.29±0.01 |
| | Cost ↓ | 11.33±3.69 | 18.60±0.35 | 2.20±2.88 | 2.57±3.23 | **0.00±0.00** | **0.00±0.00** | 1.28±1.70 | **0.00±0.00** | 0.67±1.15 |
| CarRun | Reward ↑ | **0.86±0.08** | 0.27±1.20 | **0.93±0.01** | **0.99±0.00** | 0.97±0.00 | 1.82±0.84 | **0.74±0.01** | **0.97±0.00** | 0.97±0.02 |
| | Cost ↓ | **0.83±0.76** | 30.27±10.48 | **0.20±0.18** | **0.90±0.34** | **0.11±0.17** | 24.57±20.94 | **0.00±0.00** | **0.02±0.03** | 1.92±2.49 |
| DroneRun | Reward ↑ | 0.74±0.06 | -0.18±0.09 | 0.29±0.11 | **0.58±0.00** | 0.41±0.06 | 0.50±0.08 | 0.31±0.04 | **0.36±0.12** | 0.37±0.03 |
| | Cost ↓ | 21.45±2.50 | 16.31±9.49 | 2.35±4.08 | **0.07±0.07** | 5.70±3.08 | 16.29±9.35 | 2.52±1.10 | **0.30±0.52** | 0.08±0.14 |
| AntRun | Reward ↑ | 0.81±0.05 | **0.01±0.02** | 0.03±0.05 | 0.70±0.03 | 0.53±0.13 | 0.03±0.10 | **0.43±0.02** | **0.36±0.09** | 0.30±0.20 |
| | Cost ↓ | 22.09±2.54 | **0.00±0.01** | 0.05±0.08 | 1.66±0.24 | 2.03±2.17 | **0.00±0.00** | **0.27±0.15** | 0.60±0.41 | 0.82±1.37 |
| BallCircle | Reward ↑ | 0.69±0.03 | 0.85±0.04 | 0.56±0.10 | 0.61±0.17 | **0.33±0.02** | 0.47±0.38 | 0.32±0.05 | **0.69±0.03** | 0.59±0.01 |
| | Cost ↓ | 7.55±1.95 | 10.54±1.74 | 1.11±1.92 | 2.03±1.39 | **0.01±0.02** | 10.19±17.65 | **0.00±0.00** | 0.33±0.23 | 0.00±0.00 |
| CarCircle | Reward ↑ | 0.62±0.05 | 0.74±0.06 | **0.71±0.02** | 0.71±0.01 | 0.40±0.03 | 0.70±0.01 | 0.37±0.02 | **0.69±0.01** | 0.50±0.03 |
| | Cost ↓ | 8.73±1.78 | 7.51±1.45 | **0.00±0.00** | 1.58±0.57 | 0.03±0.02 | 1.61±0.51 | **0.00±0.00** | **0.00±0.00** | 0.00±0.00 |
| DroneCircle | Reward ↑ | 0.66±0.05 | 0.86±0.07 | -0.21±0.04 | 0.55±0.01 | 0.36±0.02 | 0.43±0.04 | 0.48±0.01 | **0.53±0.02** | 0.02±0.15 |
| | Cost ↓ | 6.09±3.08 | 15.90±2.51 | 0.70±0.41 | 1.14±0.06 | **0.00±0.00** | 0.06±0.07 | 0.17±0.15 | **0.00±0.00** | 0.39±0.53 |
| AntCircle | Reward ↑ | 0.54±0.10 | 0.69±0.03 | **0.00±0.00** | 0.45±0.05 | 0.33±0.05 | 0.49±0.07 | 0.24±0.02 | **0.60±0.01** | 0.40±0.01 |
| | Cost ↓ | 9.39±3.21 | 19.03±0.84 | **0.00±0.00** | 6.59±1.42 | **0.00±0.00** | 0.02±0.03 | 0.04±0.08 | **0.02±0.03** | 0.00±0.01 |
| **Safety Gym Tasks:** $\kappa = 10$ | | | | | | | | | | |
| PointCircle1 | Reward ↑ | 0.56±0.15 | 0.74±0.15 | **0.46±0.04** | **0.54±0.01** | 0.31±0.05 | 0.57±0.08 | 0.43±0.05 | **0.52±0.01** | 0.51±0.02 |
| | Cost ↓ | 8.28±5.58 | 15.29±8.52 | **0.73±1.20** | **0.54±0.61** | 0.94±1.53 | 6.65±3.51 | | **0.04±0.04** | 0.16±0.14 |
| PointCircle2 | Reward ↑ | 0.44±0.07 | 0.74±0.07 | 0.44±0.05 | 0.59±0.02 | **0.44±0.06** | 0.03±0.75 | 0.76±0.05 | **0.55±0.01** | 0.41±0.07 |
| | Cost ↓ | 2.98±1.39 | 23.50±6.85 | 1.20±1.28 | 2.05±0.93 | **0.10±0.09** | 3.99±4.30 | 18.02±4.17 | **0.91±1.46** | 0.01±0.02 |
| PointGoal1 | Reward ↑ | 0.73±0.01 | 0.76±0.01 | 0.55±0.11 | 0.63±0.08 | 0.30±0.09 | 0.74±0.03 | 0.64±0.03 | **0.06±0.06** | 0.00±0.04 |
| | Cost ↓ | 4.03±0.22 | 3.69±0.93 | 3.33±1.68 | 2.97±0.82 | 1.45±0.74 | 3.90±0.50 | 5.38±1.05 | **0.09±0.11** | 0.10±0.11 |
| PointGoal2 | Reward ↑ | 0.68±0.02 | 0.76±0.01 | 0.25±0.13 | 0.46±0.08 | 0.19±0.04 | 0.78±0.05 | 0.31±0.07 | **0.13±0.05** | **0.14±0.06** |
| | Cost ↓ | 9.85±1.51 | 11.92±1.17 | 5.34±1.33 | 5.57±0.81 | 1.02±0.22 | 15.51±5.75 | 1.67±1.32 | 0.81±0.32 | **0.56±0.21** |
| AntVelo | Reward ↑ | 1.00±0.01 | **-1.01±0.00** | **-1.01±0.00** | 0.97±0.02 | 0.88±0.05 | **-1.01±0.00** | 0.89±0.01 | **0.99±0.01** | 0.38±0.15 |
| | Cost ↓ | 7.11±3.65 | **0.00±0.00** | **0.00±0.00** | 0.59±0.02 | 0.21±0.07 | **0.00±0.00** | **0.00±0.00** | **0.43±0.11** | 0.01±0.01 |
| HalfCheetahVelo | Reward ↑ | 1.05±0.01 | 0.93±0.03 | 0.67±0.05 | **0.98±0.00** | 0.86±0.01 | 0.91±0.04 | 0.89±0.01 | 0.96±0.01 | 0.80±0.06 |
| | Cost ↓ | 61.93±5.33 | 5.72±3.12 | 8.56±6.19 | **0.65±0.34** | 0.27±0.06 | 0.97±0.12 | **0.00±0.00** | 0.14±0.09 | 0.03±0.02 |
| SwimmerVelo | Reward ↑ | 0.51±0.08 | 0.30±0.09 | 0.20±0.20 | 0.66±0.01 | 0.45±0.06 | **0.06±0.25** | -0.02±0.05 | **0.21±0.19** | 0.20±0.12 |
| | Cost ↓ | 25.01±3.21 | 9.33±5.06 | 2.82±4.19 | 1.26±0.50 | 2.17±2.91 | **0.91±1.57** | 0.23±0.20 | **0.00±0.00** | 0.07±0.12 |

demonstrates that CARL not only enforces safety under strict budgets but also scales to more relaxed thresholds without collapsing into overly conservative behavior.

In the Safety Gym tasks, CARL's robustness is equally evident. In the particularly challenging CARCIRCLE2 task, all methods are unsafe at $\kappa = 10$ (main paper, Table 1), yet when the budget is relaxed to 40 or 80, CARL achieves both safety and competitive rewards (0.53 at cost 0.42 for $\kappa = 40$; 0.59 at cost 0.80 for $\kappa = 80$). By contrast, CAPS and CCAC remain unsafe even under the larger budgets, failing to adjust effectively. Similar improvements are visible in the other tasks, where CARL consistently satisfies safety while delivering best or competitive rewards, while both baselines oscillate between unsafe operation and lower rewards.

Overall, these results highlight that CARL is not limited to low-cost regimes: even though it employs a simple, strong penalty on unsafe transitions, it adapts gracefully as cost budgets grow. CARL demonstrates both strict constraint satisfaction in safety-critical regimes and flexibility in leveraging more permissive budgets, an essential property for practical deployment across environments with varying safety requirements.

### B.4 UNSAFE TRAJECTORIES TRAINING DATA.

Table 7 reports CARL's performance when trained either on only unsafe trajectories (*Non Safe*) or on the full dataset (*All*) ($\kappa = 5$ or 10). In both cases, CARL reliably recovers safe agents (cost $\leq 1$), but access to the full dataset generally improves the reward–cost trade-off. For example, in DRONECIRCLE and ANTCIRCLE, using all data raises rewards ($0.47 \rightarrow 0.53$ and $0.53 \rightarrow 0.60$) while further lowering costs. In ANTVELOCITY, CARL achieves nearly maximal reward (0.99) under both conditions, but with much lower cost when trained on all data (0.27 vs. 0.43). A notable case is CARCIRCLE, where CARL already finds safe solutions from unsafe-only data, yet including

Table 6: CARL results under larger cost constraints. Each cell shows the mean with standard deviation. **Bold**: safe agents ($C_{norm} \leq 1$). Gray: unsafe agents. **Blue bold**: best safe performance.

| Environment | | Cost 20 | | | Cost 40 | | |
| --- | --- | --- | --- | --- | --- | --- | --- |
| | | CAPS | CCAC | CARL | CAPS | CCAC | CARL |
| BallRun | Reward ↑ | 0.17±0.09 | **0.30±0.02** | **0.16±0.14** | 0.36±0.08 | 0.61±0.06 | **0.32±0.14** |
| | Cost ↓ | 1.41±0.79 | **0.25±0.43** | **0.44±0.45** | 1.17±0.12 | 1.13±0.13 | **0.75±0.32** |
| CarRun | Reward ↑ | **0.98±0.00** | 1.81±0.82 | **0.97±0.00** | **0.98±0.00** | 1.82±0.78 | **0.97±0.00** |
| | Cost ↓ | **0.20±0.21** | 6.39±4.79 | **0.02±0.02** | **0.21±0.26** | 3.35±2.10 | **0.03±0.02** |
| DroneRun | Reward ↑ | **0.51±0.01** | 0.46±0.11 | **0.54±0.04** | **0.51±0.04** | 0.44±0.09 | **0.52±0.05** |
| | Cost ↓ | **0.46±0.07** | 6.34±0.75 | **0.44±0.60** | **0.32±0.13** | 1.53±0.39 | **0.80±0.86** |
| AntRun | Reward ↑ | 0.65±0.03 | **0.03±0.09** | **0.55±0.06** | **0.69±0.03** | 0.05±0.07 | 0.46±0.31 |
| | Cost ↓ | 1.00±0.36 | **0.00±0.00** | **0.98±0.26** | **0.91±0.15** | 0.01±0.02 | 0.86±0.80 |
| BallCircle | Reward ↑ | **0.69±0.01** | 0.52±0.42 | 0.76±0.01 | **0.80±0.01** | 0.58±0.47 | 0.85±0.03 |
| | Cost ↓ | **0.64±0.01** | 3.19±4.53 | 1.16±0.11 | **0.82±0.01** | 1.93±1.98 | 1.12±0.09 |
| CarCircle | Reward ↑ | 0.71±0.04 | 0.72±0.01 | **0.73±0.01** | 0.75±0.01 | **0.78±0.00** | 0.74±0.02 |
| | Cost ↓ | 0.71±0.02 | 0.95±0.14 | **0.16±0.15** | 0.86±0.00 | **0.82±0.06** | 0.60±0.34 |
| DroneCircle | Reward ↑ | **0.57±0.00** | 0.40±0.07 | 0.55±0.01 | 0.62±0.00 | 0.17±0.11 | **0.65±0.03** |
| | Cost ↓ | **0.73±0.02** | 0.85±0.06 | 0.01±0.02 | 0.87±0.02 | 0.66±0.06 | **0.83±0.12** |
| AntCircle | Reward ↑ | 0.40±0.01 | 0.56±0.07 | **0.61±0.05** | 0.52±0.03 | 0.59±0.02 | **0.67±0.03** |
| | Cost ↓ | 0.09±0.05 | 0.55±0.25 | **0.02±0.02** | 0.35±0.02 | 0.95±0.23 | **0.10±0.04** |
| | | Cost 40 | | | Cost 80 | | |
| CarCircle1 | Reward ↑ | **0.47±0.03** | 0.48±0.12 | **0.43±0.02** | 0.63±0.03 | 0.46±0.10 | **0.56±0.02** |
| | Cost ↓ | **0.90±0.29** | 4.32±1.12 | **0.01±0.02** | 1.14±0.06 | 2.61±0.39 | **0.07±0.04** |
| CarCircle2 | Reward ↑ | 0.44±0.10 | 0.54±0.05 | **0.53±0.02** | 0.54±0.09 | 0.58±0.02 | **0.59±0.02** |
| | Cost ↓ | 1.08±1.03 | 2.36±2.14 | **0.42±0.17** | 1.09±0.73 | 1.82±0.58 | **0.80±0.29** |
| CarGoal1 | Reward ↑ | 0.31±0.04 | 0.84±0.02 | **0.33±0.01** | 0.40±0.06 | **0.84±0.02** | 0.41±0.06 |
| | Cost ↓ | 0.45±0.16 | 1.32±0.14 | **0.27±0.10** | 0.22±0.06 | **0.58±0.09** | 0.21±0.02 |
| CarGoal2 | Reward ↑ | **0.17±0.06** | 0.94±0.05 | **0.13±0.04** | **0.24±0.04** | 0.91±0.05 | **0.15±0.06** |
| | Cost ↓ | **0.85±0.05** | 4.31±1.17 | **0.29±0.05** | **0.35±0.10** | 2.06±0.46 | **0.18±0.07** |
| PointCircle1 | Reward ↑ | **0.51±0.02** | 0.65±0.08 | **0.55±0.02** | **0.56±0.02** | 0.68±0.06 | **0.56±0.01** |
| | Cost ↓ | **0.88±0.35** | 2.51±0.92 | **0.22±0.11** | **0.70±0.20** | 1.62±0.33 | **0.24±0.12** |
| PointCircle2 | Reward ↑ | **0.51±0.02** | -0.28±0.71 | **0.56±0.04** | 0.61±0.03 | **-0.30±0.71** | **0.60±0.01** |
| | Cost ↓ | **0.75±0.07** | 1.11±1.09 | **0.21±0.09** | 1.02±0.21 | **0.77±0.67** | **0.30±0.08** |
| PointGoal1 | Reward ↑ | **0.50±0.06** | 0.72±0.05 | **0.09±0.04** | 0.51±0.06 | **0.67±0.16** | 0.21±0.04 |
| | Cost ↓ | **0.53±0.18** | 1.12±0.16 | **0.06±0.08** | 0.34±0.06 | **0.63±0.07** | 0.04±0.02 |
| PointGoal2 | Reward ↑ | **0.34±0.06** | 0.80±0.03 | **0.15±0.01** | **0.49±0.07** | 0.78±0.08 | **0.30±0.07** |
| | Cost ↓ | **0.76±0.10** | 3.95±0.75 | **0.24±0.02** | **0.58±0.11** | 2.00±0.17 | **0.18±0.04** |

safe trajectories eliminates residual cost variance. Overall, this ablation shows that while CARL can recover safety even from unsafe-only datasets, leveraging the full dataset strengthens safety guarantees and yields the best safe performance (**bold**).

Table 7: CARL results with only unsafe trajectories data. Each cell shows reward (↑) or cost (↓). **Bold**: safe agents ($C_{norm} \leq 1$). **Blue bold**: best safe performance.

| Data | | BallCircle | CarCircle | DroneCircle | AntCircle | PointGoal1 | PointGoal2 | AntVelocity | SwimmerVelo |
| --- | --- | --- | --- | --- | --- | --- | --- | --- | --- |
| Non Safe | Reward ↑ | **0.67±0.04** | **0.69±0.01** | **0.47±0.00** | **0.53±0.08** | **0.03±0.03** | **0.13±0.06** | **0.99±0.00** | **0.17±0.02** |
| | Cost ↓ | **0.31±0.11** | **0.03±0.06** | **0.02±0.04** | **0.02±0.02** | **0.19±0.14** | **0.85±0.20** | **0.27±0.03** | **0.06±0.10** |
| All | Reward ↑ | **0.69±0.03** | **0.69±0.01** | **0.53±0.02** | **0.60±0.01** | **0.06±0.06** | **0.13±0.05** | **0.99±0.01** | **0.21±0.19** |
| | Cost ↓ | **0.33±0.23** | **0.00±0.00** | **0.00±0.00** | **0.02±0.01** | **0.09±0.11** | **0.81±0.32** | **0.43±0.11** | **0.00±0.00** |

## B.5 HARD-FILTERING TRANSITIONS

In this variant of CARL, each sampled batch is passed through the cost critic, and transitions whose predicted cost-to-go exceeds the threshold ($\kappa = 5$) are entirely removed from training. No reward relabeling is applied, so the agent learns from a reduced dataset. While this naive filtering strategy enforces a strict notion of safety during training, it discards transitions that provide valuable learning signals or partial recovery paths. As a result, policies trained with hard filtering generalize poorly and typically violate constraints at deployment, since they only optimize the original reward signal over a limited state space. Table 8 shows that this approach succeeds in only one out of eight tasks and otherwise breaches the cost threshold severely (e.g., cost 16.53 in DRONERUN, 20.52 in ANTCIRCLE). By contrast, CARL achieves safe performance across all tasks with good rewards. These results highlight that CARL's effectiveness lies not in discarding unsafe data, but in relabeling rewards with penalties, which preserves dataset coverage and enables safe, high-reward policies.

Table 8: CARL results compared to the Hard Filter variant. Each cell shows reward ($\uparrow$) or cost ($\downarrow$) with standard deviation. **Bold**: safe agents ($C_{norm} \leq 1$). Gray: unsafe agents.

| Environment | Hard Filter | | CARL | |
| --- | --- | --- | --- | --- |
| | reward $\uparrow$ | cost $\downarrow$ | reward $\uparrow$ | cost $\downarrow$ |
| BallRun | $0.58 \pm 0.46$ | $11.38 \pm 6.60$ | **$0.28 \pm 0.02$** | **$0.00 \pm 0.00$** |
| CarRun | **$0.97 \pm 0.00$** | **$0.10 \pm 0.16$** | **$0.97 \pm 0.00$** | **$0.02 \pm 0.03$** |
| DroneRun | $0.55 \pm 0.19$ | $16.53 \pm 14.32$ | **$0.36 \pm 0.12$** | **$0.30 \pm 0.52$** |
| AntRun | $0.47 \pm 0.14$ | $4.44 \pm 2.15$ | **$0.36 \pm 0.09$** | **$0.60 \pm 0.41$** |
| BallCircle | $0.70 \pm 0.02$ | $2.41 \pm 1.61$ | **$0.69 \pm 0.03$** | **$0.33 \pm 0.23$** |
| CarCircle | $0.72 \pm 0.01$ | $1.06 \pm 0.94$ | **$0.69 \pm 0.01$** | **$0.00 \pm 0.00$** |
| DroneCircle | $0.63 \pm 0.01$ | $7.79 \pm 1.16$ | **$0.53 \pm 0.02$** | **$0.00 \pm 0.00$** |
| AntCircle | $0.68 \pm 0.03$ | $20.52 \pm 2.06$ | **$0.60 \pm 0.01$** | **$0.02 \pm 0.03$** |

## B.6 EXTRA SEEDS

To further validate the statistical significance of our results, we extended the evaluation of CARL from the original three seeds (10, 20, 30) to six random seeds by adding 40, 50, and 60 for a subset of representative tasks. This expansion results in a total of 120 test trajectories per task. In addition to the standard cumulative cost metrics, we report the Safe Rate: the percentage of test episodes where the total cost remained strictly within the limit ($C \leq \kappa$).

While the primary DSRL metric focuses on average normalized cost, the safe rate provides a more granular view of reliability, distinguishing between rare catastrophic failures and consistent minor violations.

As shown in Table 9, CARL maintains high performance stability with the increased sample size. We observe that for tasks like CarCircle, DroneCircle, and SwimmerVelocity, the method achieves a perfect 100% safe rate. Even in environments with like CarRun and DroneRun, the safe rate remains effectively perfect (98%). The low standard deviations across rewards and costs further confirm that the method's performance is not an artifact of favorable seed selection.

## C EXPERIMENTAL DETAILS

### C.1 ENVIRONMENT DESCRIPTIONS

To evaluate safe offline reinforcement learning (RL) approaches, we employ a range of environments constructed on different simulation platforms, each customized for particular agents and control tasks.

**Bullet-Safety-Gym Gronauer (2022):** This benchmark suite leverages the PyBullet physics engine and features four distinct agents: Ball, Car, Drone, and Ant. It supports two main tasks: Run and Circle. In the Run scenario, agents must traverse a corridor marked by virtual safety boundaries. While these boundaries are non-colliding, crossing them results in a penalty. Speed regulation is also enforced; going beyond a specified threshold leads to further penalties. The Circle task challenges

Table 9: CARL results over 6 random seeds (120 total trajectories per task), along with the count of safe trajectories ($N_{safe}$), and the overall Safe Rate. **Bold**: safe agents ($C_{norm} \leq 1$).

| Task | Reward | Cost | # Safe | Safe Rate |
|---|---|---|---|---|
| CarRun | **0.97**±0.00 | **0.10**±0.18 | **118** | **0.98** |
| DroneRun | **0.29**±0.16 | **0.15**±0.37 | **117** | **0.98** |
| CarCircle | **0.69**±0.01 | **0.00**±0.00 | **120** | **1.00** |
| DroneCircle | **0.52**±0.03 | **0.00**±0.00 | **120** | **1.00** |
| CarGoal1 | **0.29**±0.05 | **0.87**±0.39 | **81** | **0.68** |
| PointCircle1 | **0.52**±0.01 | **0.05**±0.08 | **118** | **0.98** |
| PointGoal2 | **0.11**±0.05 | **0.74**±0.33 | **90** | **0.75** |
| AntVelocity | **0.99**±0.01 | **0.54**±0.16 | **113** | **0.94** |
| SwimmerVelocity | **0.26**±0.17 | **0.00**±0.00 | **120** | **1.00** |

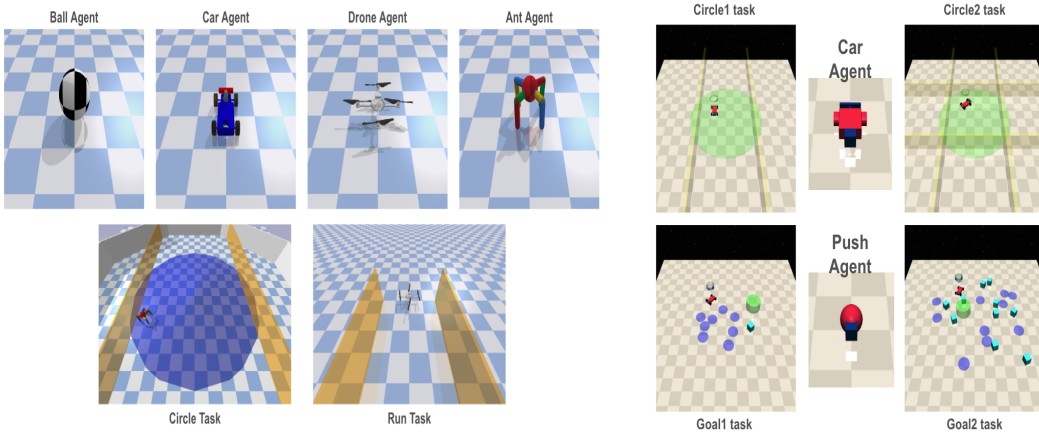

(a) Bullet-Safety-Gym environments.    (b) Safety-Gymnasium environments.

Figure 4: Visualizations of benchmark environments.

agents to move in a clockwise direction along a circular trajectory. Here, staying closer to the outer edge at higher speeds earns greater rewards, but leaving the designated safe area results in penalties. Visual examples of these scenarios are shown in Figure 4 (a).

**Safety-Gymnasium Ray et al. (2019); Ji et al. (2024):** Built on the Mujoco simulator, Safety-Gymnasium presents a diverse set of environments designed for exploring safety-related behaviors. The Car agent can engage in several tasks, including Circle and Goal, each offered in two levels of complexity. These tasks require the agent to reach specific objectives while navigating around hazards. In the Goal task, the agent moves toward designated goal positions that change after each success. The Circle task involves following a green circular track while avoiding the outer region, which incurs penalties. Figure 4 (b) displays visualizations of these environments. The benchmark also introduces velocity-restricted tasks for agents like Ant, HalfCheetah, and Swimmer. In the Velocity task, agents coordinate limb movements to advance forward while respecting velocity constraints.

## C.2 CARL HYPERPARAMETERS

Table 10 outlines the hyperparameter settings applied in various environments (BulletGym and SafetyGym) for CARL, highlighting shared configurations including cost limits, training durations, and neural network architecture details. For the IQL results, training is conducted with the IQL `tau` hyperparameter set to 0.7.

Table 10: CARL hyperparameters.

| General Parameters | BulletGym | SafetyGym |
|---|---|---|
| Cost limit | 5 | 10 |
| Training steps | 200000 | 400000 |
|    AntRun, DroneRun | 300000 | – |
| discount $\gamma$ | 0.99 | |
| Batch size | 512 | |
| Cost Critic hidden size | $256, 256$ | |
| Cost Critic learning rate | 3e-4 | |
| seeds | $10, 20, 30$ | |
| **TD3BC parameters** | | |
| Policy noise | 0.2 | |
| Policy noise clip | $(0.5, 0.5)$ | |
| Policy update frequency | 2 | |
| $\alpha$ | 2.5 | |
| Optimizer | Adam | |
| Actor, Critic learning rate | 3e-4 | |
| Actor, Critic hidden size | $256, 256$ | |

## C.3 COMPUTATIONAL RESOURCES AND TRAINING TIME:

Experiments were run on a system equipped with an NVIDIA A40 GPU (48GB memory). When running without any competing workloads, training a single task for one random seed takes approximately 31 minutes per 200k update steps.