# OpenReview forum: "Constraint-Aware Reward Relabeling for Offline Safe Reinforcement Learning"
_ICLR.cc/2026/Conference — Submitted to ICLR 2026_

### Official Review · Reviewer_tdbs · 2025-10-29

**Soundness:** 2
**Presentation:** 2
**Contribution:** 2
**Rating:** 4
**Confidence:** 4

**Summary:**

This paper proposes CARL (Constraint-aware Reward Labeling), a simple offline safe RL method that enforces safety by relabeling rewards with large penalties for unsafe state-action pairs identified through cost estimation. Experiments on DSRL tasks show that it achieves better reward performance while satisfying safety constraints under small cost budgets.

**Strengths:**

The method is simple and can be easily integrated into existing offline RL algorithms without introducing new hyperparameters or major architectural changes.

**Weaknesses:**

The proposed method lacks theoretical convergence analysis and does not provide any formal guarantee of safety during test-time deployment.

**Questions:**

1. In Eq. (2), although the constraint is defined state-wise, it remains an expectation-based formulation, effectively a soft constraint since the Q-function represents the expected cumulative cost. In contrast, FISOR formulates offline safe RL with Hamilton–Jacobi reachability constraints that treat safety violations as hard constraints. Why is the formulation in Eq. (2) better at ensuring safety than the hard-constraint formulation?

2. When relabeling the reward, $V_{max}$ represents the maximum possible infinite-horizon value. Since only an offline dataset is available, $V_{max}$ is approximated as the maximum within the dataset, which can introduce bias. How does this bias affect policy learning and safety performance, and how do the authors mitigate or correct for this bias during training?

3. Regarding the results on varying cost limits, can the proposed method generalize to new cost thresholds without retraining, as methods like CAPS and CCAC can? If retraining is required for each threshold, how do the authors ensure a fair comparison? According to Figure 2, while it is reasonable that the normalized reward increases as the cost budget increases, why does the normalized cost decrease?

---

> ### Author Response · Authors · 2025-11-21
>
> We thank the reviewer for their feedback and for taking the time to evaluate our work. We address their comments below:
>
>
> ## **Theoretical convergence analysis:**
>
> We acknowledge that formal convergence analysis for the $M=K=1$ update schedule remains an open problem. However, we argue that in the offline setting, theoretical guarantees often fail to translate into practice due to function approximation errors and optimization instability. Empirically, our results highlight this disconnect: while Lagrangian-based baselines (such as COptiDICE or BCQ-Lag) possess theoretical guarantees, they fail to satisfy safety constraints on the vast majority of benchmark tasks (e.g., COptiDICE is safe on only 2 of 19 tasks). In contrast, CARL satisfies constraints on 16 of 19 tasks. This shows that the assumptions behind the theoretical results are not robustly satisfied in practical implementations.  Thus, while we trade formal guarantees for an iterative heuristic, this trade-off yields significantly more reliable practical safety and stability in the challenging offline regime.
>
> ## **Soft constraint vs. Hamilton–Jacobi reachability:**
> We appreciate the reviewer’s comparison regarding the theoretical nature of the constraints. While we agree that Hamilton-Jacobi (HJ) reachability (used in FISOR) theoretically offers a "hard" constraint formulation, we argue that in the offline setting, the formulation in Eq. (2) (CARL) is empirically superior and more robust for three critical reasons:
> 1. **"Hard" Constraints in Offline RL:**
>
> While FISOR formulates safety as a hard constraint theoretically, practically it must approximate the feasible region using a neural network trained on static data. In offline RL, this approximation is never perfect due to function approximation errors and limited data coverage.
>
> - FISOR: If the HJ "hard" boundary is misestimated (e.g., classifying a safe state as infeasible or vice-versa due to sparse data), the policy is strictly forbidden from entering potentially recoverable regions.
> - CARL: By using $Q_c$ (which estimates the expected cost), CARL provides a dense gradient signal. Even if the estimation is imperfect, the large penalty ($r = -V_{max}$ or $-R_{max}$) naturally suppresses unsafe actions without creating artificial "hard" barriers that might shatter the optimization landscape due to estimation errors.
> 2. **Budget-Awareness vs. Structural Over-Conservatism:**
>
> The reviewer notes that FISOR enforces a hard constraint of zero violations ($V_h^*(s) \le 0$). We posit that this is actually a limitation, not a strength, for general OSRL tasks.
>
> - FISOR ignores the cost limit during training: FISOR learns a binary "feasible/infeasible" partition based on zero violation. It cannot adapt its behavior to the specific cost budget $\kappa$ provided by the user (e.g., $\kappa=10$). This leads to structural over-conservatism: even if the task allows for a small margin of error to achieve significantly higher rewards, FISOR is rigidly trained to avoid any cost, often sacrificing performance unnecessarily or failing to find a solution in sparse-reward settings.
> - CARL is budget-aware: By defining the constraint as $Q_c(s,a) \leq \kappa$ in Eq. (2), CARL directly incorporates the user's specific safety budget into the optimization landscape. This allows the policy to dynamically trade-off safety and reward up to the allowable limit, exploiting the "slack" in the constraints to maximize utility.
>
> 3. **Infeasibility due to Conflicting Constraints:**
>
> A major limitation of FISOR’s formulation is that it applies uniform constraints across the state space. In regions where the offline data is predominantly unsafe or suboptimal, FISOR attempts to simultaneously satisfy two conflicting objectives: (1) the hard safety constraint and (2) behavior regularization (staying close to the dataset distribution).
>
> - When the behavior policy itself is unsafe, these two constraints become incompatible, rendering the optimization problem infeasible. The solver effectively gets "stuck" between violating the hard constraint or deviating too far from the data support.
> - CARL avoids this conflict. By relabeling the reward to a large negative value, CARL naturally deprioritizes the behavior regularization term in the Bellman update for unsafe transitions. It does not force the agent to clone unsafe behavior, allowing it to find a "least-cost" path even when the data is imperfect.
>
> FISOR’s performance is static because its "hard constraint" training is agnostic to the test-time cost budget, whereas CARL successfully adapts its policy learning depending on the cost budget.

---

> > ### Author Response · Authors · 2025-11-21
> >
> > ## **Relabeling Bias:**
> >
> > We acknowledge the reviewer's concern that the dataset-derived $R_{max}$ might differ from the true environmental rewards. However, we argue that knowing the "true" global maximum is unnecessary for offline training.
> > 1. Self-Contained Optimization: In offline RL, the policy learns strictly from the fixed dataset. The goal of the penalty is to make unsafe actions unattractive relative to the available alternatives in that dataset. Since the agent’s value estimates are bounded by the data it has seen, a penalty calibrated to the dataset's maximum observed reward ($R_{max}$) is sufficient to flip the sign of the value estimate and discourage the action. We do not need the "true" environmental maximum because the penalty is an internal training mechanism, not a deployed reward function.
> > 2. Over-Conservatism Bias of $V_{max}$: We observed that using the theoretical infinite-horizon maximum ($V_{max} = R_{max}/(1-\gamma)$) introduces a structural "over-conservatism bias." Because $\gamma$ is close to 1, $V_{max}$ becomes extremely large, drowning out the learning signal for recoverable states. As shown in Table 3, this "harsher" penalty caused rewards in tasks such as  AntVelocity to drop precipitously (0.99 $\rightarrow$ 0.38) because the policy avoided even marginally risky but necessary transitions.
> > 3. By  setting the penalty to $R_{max}$, we provide a "less aggressive" correction that effectively prunes unsafe  transitions without suppressing the primary task signal. This approach removes the sensitivity to the discount factor $\gamma$ and provides the most robust empirical trade-off between safety and reward across the benchmark tasks.
> >
> >
> > ## **Results Fairness:**
> >
> > We clarify that CARL is a training-time method designed to solve a specific constrained optimization problem defined by a fixed $\kappa$. Unlike CAPS or CCAC, it does not support zero-shot adaptation to new thresholds without retraining. However, the comparison remains fair because we follow the standard OSRL evaluation protocol: each method is trained to solve the specific task instance defined by the budget. The experiment measures the ability of an algorithm to optimize a specific Constrained MDP (CMDP), which is the primary problem statement in the literature. While CAPS and CCAC offer the additional feature of adaptation, we compare against them to show that even without this complexity, CARL provides superior stability and performance on the core fixed-constraint tasks.
> >
> > ## **Decreasing cost:**
> >  The decrease in normalized cost as the budget increases is an artifact of the metric definition, not an indication that the agent is incurring less risk. Normalized cost is defined as $C_{norm} = C_{\pi} / \kappa$. As the budget $\kappa$ increases (e.g., from 10 to 80), the denominator grows significantly. While the agent does utilize more of the available budget (increasing absolute cost $C_{\pi}$), it generally does so conservatively, causing the ratio $C_{\pi} / \kappa$ to decrease.

---

> > > ### Comment · Reviewer_tdbs · 2025-11-25
> > >
> > > Thank you for the clarifications. However, my major concerns regarding convergence and the use of $R_{\max}$ remain. Reviewer LTcd also noted the mismatch between $V_{\max}$ in the theoretical analysis and $R_{\max}$ in the implementation. The authors explain that following the theorem and using $V_{\max}$ can lead to “over-conservatism bias”, which further highlights the gap between the theoretical guarantees and the practical algorithm. Therefore, I maintain my score.

---

### Official Review · Reviewer_EjRo · 2025-10-30

**Soundness:** 1
**Presentation:** 2
**Contribution:** 1
**Rating:** 2
**Confidence:** 5

**Summary:**

This paper proposes a minimal approach, named Constraint aware Reward (Re)Labeling (CARL), to translate offline safe RL to offline RL via reward shielding (relabeling).

CARL contains the following content:

1. **Iterative Policy Improvement**
   1. offline policy evaluation $\to$ reward relabeling $\to$ offline policy optimization.
2. **Batch Reward Relabeling**
   1. Relabel the reward, which violates the constraint to a large negative value.

**Strengths:**

This paper has done comprehensive experiments on many benchmarks compared with latest baselines.

This paper writes a detailed related work on offline safe RL.

This paper translates the constrained optimization problem into a shielding optimization problem, where the reward relabeling can be regarded as an improved shielding method in Safe RL.

**Weaknesses:**

### The key idea of this paper shares a very similar essence to CPQ [1]. However, the main difference between CPQ and CARL, and the improvement/superiority of CARL over CPQ are not explained clearly. The contribution may be limited before clarifying the following issue:

1. Both CPQ and CARL share the same **Sketch of an Iterative Policy Improvement Algorithm** in Eq. (4) of this paper.
   1. In CPQ, it follows: offline policy evaluation $\to$ reward value relabeling $\to$ offline policy optimization.
   2. That is to say, CPQ already utilizes this approach in 2022.

2. **The only difference is that CPQ relabels the value to 0, while CARL relabels the reward to a large negative value.**
   1. CPQ utilizes value function relabeling:
      1. $Q_r=1_{\{Q_c\leq k\}}\cdot Q_r'$
      2. This is to say that CPQ relabels the unsafe condition as 0.
   2. While CARL utilizes reward relabeling in Eq. (3) and (5).
      1. $r_\pi(s,a)=1_{\{Q_c\leq k\}}\cdot r(s,a)-1_{\{Q_c> k\}}\cdot V_{max}$
      2. This is to say that CARL relabels the unsafe condition as $-V_{max}$.
   3. Consider the value function is a cumulation of rewards; there seems to be no large difference between relabeling rewards and relabeling the value function directly.
      1. Originally, the reward value follows: $Q_r = r+\gamma Q_r'$
      2. In CPQ, the reward value follows: $Q_r = r+\gamma \cdot 1_{\{Q_c\leq k\}}\cdot Q_r'$, which is
         1. $Q_r = r+\gamma \cdot 1\cdot Q_r'$ for $Q_c\leq k$
         2. $Q_r = r+\gamma \cdot 0\cdot  Q_r'$ for $Q_c> k$
      3. In CARL, the reward value follows:
         1. $Q_r = r\cdot 1+\gamma \cdot Q_r'$ for $Q_c\leq k$
         2. $Q_r = -V_{max}+\gamma \cdot  Q_r'$ for $Q_c> k$
      4. Considering that the rewards are usually positive values (Even if they are not, the rewards can be regularized to be positive without loss of generality
         1. **Both CPQ and CARL aim to set the reward value that violates the constraint to a small value.**
         2. Is there an essential difference between relabeling to 0 or $-V_{max}$?
         3. Is there an essential difference between relabeling the reward signal or the reward value function?
   4. **This difference is hard to be regarded as the central contribution of a paper if it is not explained or compared clearly.**
3. CPQ naturally follows $K=M=1$, where the OPE and OPO are jointly optimized.
   1. What is the necessity of discussing the content of this part if this finding is already utilized in previous work?
   2. CPQ also suffers from the oscillation in Figure 1 even if $K=M=1$. Is there any explanation about the problem? Or are $K$ and $M$ really the main reason for this problem?
4. CPQ shares all **Summary of CARL’s advantages** in line 309, page 6, as they share a similar essence:
   1. CPQ can also be wrapped around existing offline RL algorithms.
   2. CPQ also doesn’t introduce any additional hyperparameters. (The additional hyperparameters in CPQ are related to another OOD action problem, which is not considered in CARL.)
   3. The main reason is that the only (or main) difference between CPQ and CARL is the above relabeling methods.

5. Besides, CPQ additionally addresses the OOD action problem. However, the performance of CARL is much better to CPQ.
   1. Why does CARL not consider the OOD action problem, considering that the Bellman backup procedure is also utilized in CARL?

   2. Why CARL works much better than CPQ?

6. It is very important to clarify that from Section 4 to 5,
   1. which content is already proposed in previous work (like CPQ)?
   2. which content is completely innovative part of CARL?
   3. why and how do the innovative parts improve CARL against CPQ?
   4. where does the performance gain come from?



### Although there is no doubt that TD3BC can be applied to CARL, there are still problems about how to apply IQL to CARL, which is glossed over in this paper.

1. TD3BC has explicit policy during the training procedure. Thus, CARL can estimate $Q^\pi_r$ and $Q^\pi_c$ under the same policy.
   1. The OOD problem can be mitigate by BC but can not be avoided as long as the Bellman backup procedure is updated with explicit policy.
2. To avoid the OOD problem completely, IQL propose to update the $Q^\pi_r$ implicitly via expectile regression, where the policy is implicitly hidden in the expectile regression to avoid explicitly sampling action during Bellman backup procedure.
3. Considering this implicit design, it is impossible to estimate $Q^\pi_r$ and $Q^\pi_c$ under the same policy without extracting the policy out.
   1. Think about why IQL regards "value function update" and "policy extraction" as two separate procedure.
      1. To avoid the OOD problem, the value function update cannot utilize the extracted policy.
4. This is why C2IQL [2] is proposed to to apply IQL's idea in Offline Safe RL to address this problem.
   1. C2IQL tries to combine CPQ and IQL without breaking the implicit property to avoid OOD problem.
   2. Since CPQ is similar to CARL, C2IQL is also similar to IQL+CARL, while C2IQL additionally points out and addresses this problem
   3. To understand the above content, please understand the idea of IQL, IDQL, C2IQL instead of merely treating them as an algorithm.
5. If this paper utilize the extracted policy to estimate the $Q^\pi_c$, it has no difference between TD3BC and IQL when applied to CARL.
   1. Extracted policy breaks IQL's key idea. When this policy is utilized to influence value function update, IQL has no difference from other actor-critic method.
   2. Thus the backbone analysis in line 369, page 7 seems to be not very meaningful.
6. If this paper utilize framework that implicitly keeping both functions under the same policy like C2IQL, please explain the design in detail with proper citation.
7. Considering the similarity between C2IQL (CPQ+IQL) and CARL+IQL, please verify from section 4 and 5 again:
   1. which content is already proposed in previous work;
   2. which content is completely innovative part of CARL;
   3. why and how does the innovative parts improve CARL previous work;
   4. where does the performance gain come from;

[1] Xu, H., Zhan, X., & Zhu, X. (2022, June). Constraints penalized q-learning for safe offline reinforcement learning. In *Proceedings of the AAAI Conference on Artificial Intelligence* (Vol. 36, No. 8, pp. 8753-8760).

[2] Zifan, L. I. U., Li, X., & Zhang, J. C2IQL: Constraint-Conditioned Implicit Q-learning for Safe Offline Reinforcement Learning. In *Forty-second International Conference on Machine Learning*.

**We highly hope that this paper can clarify these issues, as they may potentially lead to academic plagiarism in severe cases.**

**Questions:**

Please see the Weakness part.

---

> ### Author Response · Authors · 2025-11-21
>
> We thank the reviewer for their feedback and for taking the time to evaluate our work. We address their comments below:
>
> ## **CPQ vs CARL:**
> First, the statement “Both CPQ and CARL share the same Sketch of an Iterative Policy Improvement Algorithm in Eq. (4) of this paper” is an oversimplification of nearly all OSRL methods. By design, OSRL algorithms almost always follow an iterative policy improvement process augmented with a cost signal (e.g., the standard Lagrangian formulation falls exactly into this paradigm). Therefore, asserting that CPQ and CARL share the same sketch is far too coarse. What matters is how the cost signal is incorporated into the value and/or policy update. This is precisely where CARL fundamentally differs from CPQ.
>
> To compare CPQ and CARL at the level of algorithmic principles, we will intentionally ignore all other differences between the two methods (e.g., critic updates, policy extraction mechanisms), and focus only on the key principles: **CPQ sets Q_r to 0 when Q_c​ indicates a safety violation**, whereas **CARL penalizes by assigning a negative reward when Q_c​ indicates a violation**.
>
> Although this may appear to be a small change, it leads to a fundamental difference, as demonstrated empirically in our paper, and in the additional experiments we provide in this rebuttal and as elaborated below.
>
> At first glance, CPQ’s logic seems natural and even ideal: if an action is deemed unsafe according to Q_c​, we prevent selecting it by setting its value to zero. However, this approach is overly conservative. Since the cost estimator Q_c​ can be inaccurate, especially early in training, CPQ prematurely marks actions as unsafe by zeroing out their value. Concretely, consider an action a flagged as unsafe by Q_c​ during the initial iterations. CPQ immediately enforces Q_r(s,a) = 0, whereas CARL assigns a negative reward to (s,a). Later, as more iterations pass and Q_c​ becomes more accurate, we may discover that action a is actually safe and leads to high reward. Yet, because CPQ has already locked in Q_r(s,a)=0, it’s hard to correct this early error, CPQ continues to ignore the future high-reward trajectory stemming from (s,a). CARL, by contrast, only penalizes at the per-step reward level, not at the trajectory level, which allows these errors to be corrected naturally as learning proceeds.
>
> Our experiments clearly illustrate the advantages of CARL over CPQ. In Table 3, CARL with an ⁡R_{\max}​ penalty outperforms CARL with a ⁡V_{\max} penalty, showing that stronger penalties can be detrimental. But even the conservative V_{\max}​ variant of CARL significantly outperforms CPQ, reinforcing that CPQ’s overly conservative penalization mechanism is not beneficial for OSRL. Per-step reward penalization in CARL—even in conservative form—is more appropriate and more effective than trajectory-level penalization in CPQ.
>
> Finally, we present additional experiments to show that CARL’s superior performance is driven by its core reward relabeling mechanism rather than architectural factors such as a stronger backbone or the removal of the VAE. These ablation studies isolate the safety mechanism from the model architecture and demonstrate that CPQ’s value masking is insufficient for offline safety, whereas CARL’s penalty relabeling provides substantially more robust behavior.
>
> ## **Experiments:**
>
> - **Variant 1: Value Masking (CPQ-style Critic):** In this variant, we replaced CARL's reward relabeling with CPQ's critic update logic: for unsafe transitions, we kept the immediate reward $r$ but set the future bootstrap value to zero.
>
> Result: The agents were unsafe. This indicates that retaining the immediate reward $r$ leaves unsafe transitions attractive to the agent, effectively luring it into violation despite the zeroed future value.
> | Task | Reward  | Cost |
> |-|-|-|
> | BallRun | 0.91 ± 0.11   | 17.10 ± 1.21   |
> | CarRun | 0.99 ± 0.03   | 13.84 ± 12.94  |
> | DroneRun  | 0.87 ± 0.07   | 23.30 ± 3.54   |
> | AntRun  | 0.40 ± 0.26   | 7.69 ± 6.71 |
> | BallCircle  | 0.89 ± 0.01   | 11.40 ± 0.39   |
> | CarCircle | 0.88 ± 0.01   | 16.78 ± 0.31   |
> | DroneCircle  | 0.93 ± 0.00   | 19.14 ± 0.21   |
> | AntCircle | 0.92 ± 0.07   | 29.74 ± 2.87   |
>
> - **Variant 2: Policy Filtering (CPQ-style Actor):** Building on Variant 1, we added CPQ's policy extraction mechanism, which explicitly filters out unsafe actions during the maximization step.
>
> Result: The method failed to consistently recover safe policies. This indicates that masking creates a "blind spot" with zero gradients in unsafe regions, leaving the policy with no directional signal to recover; unlike CARL’s repulsive penalty which actively pushes the agent toward safety, masking simply silences the learning signal, causing the agent to get stuck or drift into unsafe territory without a correction mechanism.

---

> > ### Author Response · Authors · 2025-11-21
> >
> > | Task   | Reward        | Cost            |
> > |---------------|---------------|-----------------|
> > | BallRun       | 1.19 ± 0.01   | 18.13 ± 0.12    |
> > | CarRun        | -0.54 ± 0.31  | 33.97 ± 2.49    |
> > | DroneRun      | 0.16 ± 0.36   | 15.56 ± 6.52    |
> > | AntRun        | 0.43 ± 0.12   | 15.58 ± 1.96    |
> > | BallCircle    | 0.21 ± 0.32   | 27.03 ± 9.15    |
> > | CarCircle     | 0.25 ± 0.05   | 22.15 ± 2.88    |
> > | DroneCircle   | 0.09 ± 0.08   | 17.14 ± 4.28    |
> > | AntCircle     | 0.30 ± 0.11   | 38.67 ± 1.91    |
> >
> > The failure of these variants demonstrates  that Reward Relabeling ($r \to -V_{max}$) is more robust than Value Masking ($Q \to r$). By overwriting the immediate reward with a large negative penalty, CARL transforms the optimization landscape itself, creating a deep "valley" at unsafe transitions. This ensures that both the immediate and future value estimates are maximally negative, naturally aligning the critic's gradients with the safety constraint. In contrast, CPQ's logic creates a conflict where the critic may still value unsafe actions (due to positive $r$), requiring an explicit and often insufficient "hard filter" to prevent the agent from selecting them.
> >
> >
> > - **CPQ is a wrapper / No Hyperparams:**
> >
> > CPQ is an architectural modification that requires implementing specific, custom loss functions for both the reward and cost critics (Eq. 1 and 3 in the CPQ) and training a conditional VAE for OOD detection. Conversely, CARL acts as a true data-level wrapper that preprocesses the batch rewards before passing them to any off-the-shelf algorithm (like IQL or TD3-BC) without altering the underlying code or loss functions, using a fixed schedule ($M=K=1$) that requires no tuning.
> >
> > - **Why CARL ignores OOD:**
> >
> > CARL does not ignore the OOD problem; rather, it strategically offloads it to state-of-the-art backbone offline RL algorithms (such as TD3-BC and IQL) which have built-in mechanisms for handling distribution shifts (e.g., behavior cloning regularization or expectile regression). CPQ, by contrast, attempts to solve OOD explicitly through its own VAE-based penalty, which effectively couples its safety mechanism to the quality of its generative model. CARL’s performance gain stems from decoupling these problems: it uses aggressive reward relabeling for safety and leverages the superior, modern OOD regularization of its backbone offline RL approach for stability.
> >
> > - **Regarding Oscillation:**
> >
> > It is important to distinguish between the sources of instability in these two contexts. The oscillation shown in Figure 1 is caused explicitly by update drift: the cost-critic $Q_c$ becomes desynchronized from the current policy $\pi$, leading to inaccurate penalty assignments. This specific form of instability is directly resolved by synchronizing updates ($K=M=1$). Regarding the reviewer's observation that CPQ also suffers from oscillation: Since CPQ inherently utilizes a synchronized update scheme, its instability is fundamentally different from the "drift" phenomenon illustrated in Figure 1. While Figure 1 isolates an issue of update frequency specific to our wrapper design, the oscillation in CPQ points to structural issues within that specific method (e.g.,  filtered policy extraction, VAE instability) rather than the update schedule itself.
> >
> > - **IQL Integration:**
> >
> > Our use of IQL within CARL preserves the core implicit-policy structure of IQL while enabling the cost-critic updates required by CARL. In CARL, the backbone offline RL algorithm receives only relabeled rewards and is never exposed to Qc or to the original rewards. Therefore, the IQL reward critic continues to be trained exactly as in standard IQL, via expectile regression on dataset actions only, without querying the current policy in Bellman backups and without introducing additional OOD actions. The only component that requires access to the extracted policy is the cost-critic Qc, which is evaluated through standard off-policy evaluation (OPE) of the current policy \pi. This OPE step does not alter the IQL value-learning mechanism, since no policy-sampled actions are fed back into the reward critic or value function updates. Thus, CARL maintains the separation that IQL relies on, implicit value learning for rewards and explicit policy extraction, while attaching a cost-critic whose role is limited to determining when to relabel a transition as unsafe. In this sense, CARL does not convert IQL into an explicit actor–critic method; instead, it uses IQL unchanged on the reward side and relies on a separate, OPE module to obtain the cost estimates needed for CARL’s relabeling rule. This design keeps the backbone offline RL method fully intact while enabling CARL to enforce safety constraints in a way that is agnostic to whether the backbone is explicit (TD3-BC) or implicit (IQL).

---

> > > ### Author Response · Authors · 2025-11-21
> > >
> > > - **C2IQL similarity:**
> > >
> > > C2IQL mathematically couples the cost and reward updates, deriving the cost value function $V_c$ using the same expectile weights ($M$) as the reward function. While theoretically elegant, this rigidly ties the safety mechanism to the specific mathematics of IQL. In contrast, CARL is designed as a universal wrapper. By keeping the cost evaluation ($Q_c$) explicit and separate from the backbone, CARL can be applied to any offline RL algorithm (TD3-BC, IQL, DT) without deriving new math for every backbone.

---

> > > > ### Comment · Reviewer_EjRo · 2025-11-27
> > > >
> > > > ### **Q1: I sincerely hope that the authors will not have such great hostility towards me and give me lots of answers irrelevant to my question. If the authors think my review is not reasonable, it is highly recommended that other reviewers, AC, and everyone to look at my opinions and think its rationality.**
> > > >
> > > > 1. If the authors mentioned all contents in next question (Q2) and introduced CARL as an incremental paper on existing works in the initial draft, I would like to give a score of 4 and may improve my score if the authors addressed all my concerns. However, this paper even not mention this similarity, where I think that **it may potentially lead to academic plagiarism in severe cases**.
> > > >
> > > > 2. The authors' answers are emotional and the answer is irrelevant to the question.
> > > >    1. All my opinions have list lots of details and reasons for why. However, the authors seem to answer none of them.
> > > >    2. The author's response completely distorted my opinion, for example:
> > > >       1. First, the statement “Both CPQ and CARL share the same Sketch of an Iterative Policy Improvement Algorithm in Eq. (4) of this paper” is an oversimplification of nearly all OSRL methods. By design, OSRL algorithms almost always follow an iterative policy improvement process augmented with a cost signal (e.g., the standard Lagrangian formulation falls exactly into this paradigm).
> > > >       2. Second, my question on IQL backbone.
> > > >       3. Detailed explanation will be shown in the following questions.
> > > >
> > > > ### **Q2: The authors also agree that CPQ and CARL share similarity and small changes on algorithmic level. However, the authors never mention, compare, and analyze the difference in the main content of the paper even if I have mentioned this point.**
> > > >
> > > > 1. I sincerely hope the authors can answer my following questions.
> > > >    1. CPQ was an well-known algorithm designed 3 years ago. In this paper, CPQ is also included as a baseline for comparison. Do the authors notice the similarity between CPQ and CARL?
> > > >    2. If the authors noticed the similarity, why didn't they compare the similarity?
> > > >    3. If the authors didn't notice the similarity, did they read CPQ's paper carefully before including it as  a baseline, and why didn't they compare the similarity after I mentioned this point?
> > > >    4. Compared with CPQ and C2IQL, do the authors think CARL is an innovative and novel framework or an incremental paper?
> > > >
> > > > ### **Q3: The author's response completely distorted my opinion or did not attempt to understand what I was saying.**
> > > >
> > > > 1. The authors said my first state "Both CPQ and CARL share the same Sketch of an Iterative Policy Improvement Algorithm in Eq. (4) of this paper" is an oversimplification of nearly all OSRL methods.
> > > >    1. The reason is "OSRL algorithms almost always follow an iterative policy improvement process augmented with a cost signal"
> > > >    2. This reason completely distorted my opinion.
> > > > 2. The Iterative Policy Improvement Algorithm in Eq. (4) is:
> > > >    1. offline policy evaluation $\to$ reward relabeling $\to$ offline policy optimization
> > > >    2. The reward relabeling part is very important to make sure the policy can satisfy the constraint.
> > > > 3. Most OSRL algorithms follows:
> > > >    1. offline policy evaluation $\to$ Lagrangian formulation $\to$ offline policy optimization
> > > >    2. The Lagrangian formulation is a different way compared with reward relabeling on the following reason:
> > > >       1. Lagrangian formulation will not change the value of the reward/cost value function directly
> > > >       2. Lagrangian formulation will not change the dataset values.
> > > >       3. Lagrangian formulation only influence policy optimization by gradually changing the Lagrangian multiplier.
> > > > 4. As far as I know, OSRL methods related to Eq. (4) only involves:
> > > >    1. CPQ, 2022,
> > > >    2. C2IQL, July, 2025,
> > > >    3. CARL in this paper.
> > > > 5. I cannot understand the reasons given by the authors, nor can I understand why they distort my opinions.

---

> > > > > ### Comment · Reviewer_EjRo · 2025-11-27
> > > > >
> > > > > ### Q4: I have listed the updating formula of CPQ and CARL step by step and compared step by step to show the similarity **on the core idea** in my first review. However, the authors' answer completely ignores what I mentioned and discusses how CPQ and CARL are different **on experiment results and detailed implementation**.
> > > > >
> > > > > 1. Similarity on masked/relabeled value:
> > > > >    1. Both CPQ and CARL propose to mask/relabel unsafe values to be small
> > > > >       1. CPQ masked to 0.
> > > > >       2. CARL masked to a negative value.
> > > > >    2. **Since nearly all reward values are positive numbers in all tested benchmarks, what is the difference when comparing the negative value with 0?**
> > > > > 2. Similarity in reward value function update:
> > > > >    1. CPQ and CARL treat $Q_c\leq k$ as the same case:
> > > > >       1. CPQ:   $Q_r = r+\gamma \cdot 1\cdot Q_r'=r+\gamma\cdot Q_r'$ for $Q_c\leq k$
> > > > >       2. CARL: $Q_r = r\cdot 1+\gamma \cdot Q_r'=r+\gamma\cdot Q_r'$ for $Q_c\leq k$
> > > > >    2. CPQ and CARL treat $Q_c> k$ similarly:
> > > > >       1. CPQ:
> > > > >          1. $Q_r = r+\gamma \cdot 0\cdot  Q_r'$ for $Q_c> k$
> > > > >          2. The final result is $Q_r$ that violates the constraint is assigned to a small value.
> > > > >       2. CARL:
> > > > >          1. $Q_r = -V_{max}+\gamma \cdot  Q_r'$ for $Q_c> k$
> > > > >          2. The final result is $Q_r$ that violates the constraint is assigned to a negative value.
> > > > >    3. What is the difference **on the key idea**?
> > > > >       1. I want to know the theoretical difference **on the key idea**, not the simple difference on the surface.
> > > > > 3. Similarity in policy optimization:
> > > > >    1. CPQ:
> > > > >       1. $\max_\pi\ \ 1(Q^c\leq k)\cdot Q^r$, where $Q^r$ is masked.
> > > > >    2. CARL:
> > > > >       1. $\max_\pi\ \ Q^r$, where $Q^r$ is relabeled.
> > > > >
> > > > > 4. **A single coincidence can be dismissed as just that. However, if coincidences keep appearing, can we still regard them as mere coincidences?**
> > > > > 5. **Does the author really think the above differences can be the innovative and novel contribution of this paper?**
> > > > >
> > > > > ### **Q5: The author says: CARL, by contrast, only penalizes at the per-step reward level, not at the trajectory level, which allows these errors to be corrected naturally as learning proceeds. Here is my confusion:**
> > > > >
> > > > > 1. CPQ and CARL treat $Q_c> k$ similarly:
> > > > >    1. CPQ:
> > > > >       1. $Q_r = r+\gamma \cdot 0\cdot  Q_r'$ for $Q_c> k$
> > > > >       2. The final result is $Q_r$ that violates the constraint is assigned to a small value.
> > > > >    2. CARL:
> > > > >       1. $Q_r = -V_{max}+\gamma \cdot  Q_r'$ for $Q_c> k$
> > > > >       2. The final result is $Q_r$ that violates the constraint is assigned to a negative value.
> > > > >
> > > > > 2. The authors claim that CARL only penalizes at the per-step reward level.
> > > > >    1. What is the relabeled reward used for?
> > > > >       1. Is it used to update the reward value function?
> > > > >       2. Is the updated reward value function used as the Bellman backups when updating the reward values of other state-action pairs?
> > > > >    2. In the Bellman backup procedure,
> > > > >       1. CARL changes the first term.
> > > > >       2. CPQ changes the second term.
> > > > >       3. Why does CARL penalize on the step level while CPQ penalizes on the trajectory level?
> > > > >
> > > > > ### **Q6: About CARL, IQL, C2IQL. The authors continually explain that the reward critic and policy did not introduce additional OOD actions. However, the authors never explain whether the cost critic introduces additional OOD actions or not.**
> > > > >
> > > > > 1. Please give me the clear update formula and equation on the IQL-based CARL.
> > > > > 2. We agree with the authors' explanation of "Thus, CARL maintains the separation that IQL relies on, implicit value learning for rewards and explicit policy extraction."
> > > > >    1. We also agree that the reward critic does not introduce additional OOD actions.
> > > > > 3. The question part is whether the cost-critic $Q_c$ is updated in AC style or in IQL style.
> > > > >    1. If $Q_c$ is updated in AC style (the author mentions that "Qc is evaluated through standard off-policy evaluation (OPE) of the current policy $\pi$."), then $Q_c$ must query the current policy in Bellman backups and thus introduces additional OOD problems.
> > > > >       1. This is why I say "If this paper utilizes the extracted policy to estimate the $Q_c$, it has no difference between TD3BC and IQL when applied to CARL," in my first review.
> > > > >    2. If $Q_c$ is updated in IQL style, then how to keep $Q_c$ under policy $\pi$ becomes an important problem, which is the main problem that C2IQL tries to address.
> > > > >
> > > > > 4. The author's explanation of C2IQL also clearly shows that the author did not try to understand how IQL-style algorithms are designed.
> > > > >    1. The reason why C2IQL is mentioned is not to show that CARL is less superior than C2IQL, but to show why this "rigid tie" is necessary in IQL-style algorithms. This can help the authors understand the key idea of IQL-style algorithms.

---

> > > > > > ### Author Response · Authors · 2025-12-04
> > > > > >
> > > > > > We must **firmly object** to the tone and language used in this review. Accusations of *“hostility”* and *“academic plagiarism”* are serious, defamatory, and entirely unsubstantiated. Furthermore, the claim that we did not read a baseline paper—one which we explicitly cited, analyzed, and performed specific ablation studies against in our rebuttal—is factually incorrect and dismissive of the scientific effort we have provided.
> > > > > >
> > > > > > We respectfully request that the discussion maintains a professional standard, focusing strictly on the empirical evidence and mathematical definitions rather than personal characterizations or distortions of our previous responses.
> > > > > >
> > > > > > ---
> > > > > >
> > > > > > ## Q2:
> > > > > >
> > > > > > The reviewer initially acknowledged that *“This paper writes a detailed related work on offline safe RL.”*
> > > > > > In that section, we categorized methods based on their primary mechanisms. CPQ is **not** primarily a reward relabeling method; it relies on VAE-based OOD detection, cost overestimation, and policy masking. Since CARL is a *pure* reward relabeling method, we focused our textual comparison on other relabeling techniques. However, we certainly recognize CPQ as a key baseline, which is why it was included in our experiments.
> > > > > >
> > > > > > To the extent that structural comparison needed more detail—as per the review comments—we have addressed this through new ablations in the rebuttal. This is the purpose of the review process: to refine and improve the paper.
> > > > > >
> > > > > > We explicitly state in the paper that CARL is an *“embarrassingly simple framework.”* However, we strongly reject the notion that it is merely “incremental” compared to CPQ or C2IQL for the following reasons:
> > > > > >
> > > > > > 1. **The reviewer suggests that the difference between CPQ and CARL is trivial.**  If that were true, replacing CARL’s relabeling with CPQ’s masking (while keeping everything else constant) should yield similar results. It does **not**. In our Rebuttal Experiments (Variant 1 & 2), CPQ-style masking caused the agent to **fail to learn safe policies**, while CARL succeeded. CARL achieves safety on **16/19** tasks, whereas CPQ achieves safety on only **7/19** tasks, often with lower reward. A method that succeeds across the majority of benchmark tasks where the “predecessor” fails cannot be dismissed as a minor increment.
> > > > > >
> > > > > > 2. C2IQL is rigidly tied to the IQL method (coupling cost and reward expectiles). CARL is a universal wrapper that is algorithm-agnostic. CARL is minimalist, whereas C2IQL introduces cost reconstruction and constraint conditioning. They represent fundamentally different design philosophies.
> > > > > >
> > > > > > ---
> > > > > >
> > > > > > ## Q3:
> > > > > >
> > > > > > The reviewer states that we “distorted” their opinion regarding Eq. (4) and argues that only CPQ, C2IQL, and CARL follow the structure *Evaluation → Relabeling → Optimization*, whereas Lagrangian methods do not.
> > > > > >
> > > > > > We respectfully disagree with this taxonomy for several reasons:
> > > > > >
> > > > > > ### 1. **CPQ is not a “reward relabeling” method.**
> > > > > > As previously explained, CPQ does *not* simply relabel reward values. Instead:
> > > > > >
> > > > > > - It masks the bootstrap value (nullifies future \(Q_r\)).
> > > > > > - It masks policy actions on unsafe transitions.
> > > > > > - It uses a VAE for OOD detection.
> > > > > > - It uses a Lagrangian multiplier to increase costs for OOD samples.
> > > > > >
> > > > > > Classifying CPQ as similar to CARL, which purely modifies the scalar reward signal \(r\), is technically inaccurate.
> > > > > >
> > > > > > ### 2. **Lagrangian methods *do* modify the optimization signal, just not the dataset.**
> > > > > > The reviewer argues that Lagrangian methods differ because they do not change dataset values. But functionally, both approaches modify the stream used in the update:
> > > > > >
> > > > > > - CARL: modifies the reward stream → \(r \to -V_{\max}\)
> > > > > > - Lagrangian: modifies the loss signal → \(r \to r - \lambda c\)
> > > > > >
> > > > > > Neither permanently alters the static dataset; both dynamically alter the optimization landscape.
> > > > > >
> > > > > > ### 3. **CARL is not a minor variation of CPQ**
> > > > > > The reviewer characterizes the difference between CPQ and CARL as a 'minor variation' of a shared sketch. We argue that the significance of an algorithmic change is ultimately defined by its impact. CPQ’s design (VAEs, masking) achieves safety on only 7/19 tasks. In contrast, CARL’s simpler reward relabeling mechanism achieves safety on 16/19 tasks. Even if one were to concede the premise of relabeling similarity, which we dispute given the discussions above, a modification that yields a greater than 2x improvement in safety success rate represents a significant advancement, not a trivial variation.
> > > > > >
> > > > > > ---
> > > > > >
> > > > > > ## Q4
> > > > > >
> > > > > > We appreciate the reviewer’s breakdown of update rules, but must point out a critical inaccuracy regarding how CPQ and CARL handle unsafe actions.
> > > > > >
> > > > > > These differences are **not superficial**; they represent **fundamentally different theoretical behaviors** of the Bellman operator.

---

> > > > > > > ### Author Response · Authors · 2025-12-04
> > > > > > >
> > > > > > > ### 1. **Difference in Value Update**
> > > > > > >
> > > > > > > The reviewer asks: *“What is the difference between masking to 0 (CPQ) and masking to a negative value (CARL)?”*
> > > > > > >
> > > > > > > The comparison misses the role of the immediate reward \(r\).
> > > > > > >
> > > > > > > #### **CPQ Update (Bootstrap Masking)**
> > > > > > >
> > > > > > > For unsafe transitions:
> > > > > > >
> > > > > > > $y = r + 0$
> > > > > > >
> > > > > > > If $r$ is positive, the target remains positive. Thus the agent may still be drawn toward unsafe actions because the immediate reward is preserved.
> > > > > > >
> > > > > > > #### **CARL Update (Reward Overwrite)**
> > > > > > >
> > > > > > >
> > > > > > > $y = -V_{\max} + \gamma Q_r(s',a')$
> > > > > > >
> > > > > > >
> > > > > > > The immediate reward is overwritten with a large negative penalty.  The entire value becomes negative, actively repelling the agent from unsafe states.
> > > > > > >
> > > > > > > **Summary:**
> > > > > > > - CPQ cuts off the future, leaving “reward traps.”
> > > > > > > - CARL penalizes the action directly, creating consistent gradients toward safe behavior.
> > > > > > >
> > > > > > > ### 2. **Difference in Policy Optimization**
> > > > > > >
> > > > > > > - **CPQ:** Uses an indicator $\mathbb{1}(Q_c \le l) $ inside the maximization → *hard mask*, zero gradient in unsafe regions.
> > > > > > > - **CARL:** Uses standard maximization over shaped rewards → non-zero gradients even in unsafe regions, enabling recovery.
> > > > > > >
> > > > > > > Our ablations show that forcing CARL to use CPQ-style masking breaks performance entirely.
> > > > > > >
> > > > > > >
> > > > > > > ---
> > > > > > >
> > > > > > > ## Q5
> > > > > > >
> > > > > > > In CARL, the relabeled reward is used to update the reward value function, and the updated reward value function is used for bootstrapping in subsequent backups ($\gamma V(s')$).
> > > > > > >
> > > > > > > The reviewer asks why we distinguish between penalizing the "first term" (CARL) and the "second term" (CPQ). This mathematical distinction dictates the optimization landscape:
> > > > > > >
> > > > > > > ### CPQ (Trajectory-Level / Bootstrap Modification)
> > > > > > > As defined in the CPQ paper, the operator is:
> > > > > > > $$
> > > > > > > \mathcal{T}_{P}^{\pi}Q(s,a) = r + \gamma \mathbb{E}[\mathbb{1}(\text{safe}) V(s')]
> > > > > > > $$
> > > > > > > For an unsafe transition, this effectively sets $Q(s,a) = r + 0$.
> > > > > > >
> > > > > > > CPQ retains the immediate reward $r$ (the step) but masks the bootstrap $\gamma V(s')$ (the future trajectory). If an action yields a high immediate reward $r$, the target value $r+0$ remains positive. This creates a conflicting signal: the agent is penalized by the loss of future value but still incentivized by the immediate reward. We refer to this as "trajectory-level" because it penalizes by discarding the future trajectory expectation.
> > > > > > >
> > > > > > > ### CARL (Step-Level / Reward Modification)
> > > > > > >
> > > > > > >
> > > > > > > CARL updates the value target as: $Q(s,a) = -V_{\max} + \gamma V(s')$
> > > > > > >
> > > > > > > CARL explicitly penalizes the immediate step ($r \to -V_{\max}$) but preserves the bootstrap term $\gamma V(s')$. The large negative constant $-V_{\max}$ dominates the equation, ensuring that the total Q-value is negative. This creates a consistent penalty gradient that discourages the action immediately, while the preservation of $\gamma V(s')$ ensures that the value function remains consistent for the rest of the trajectory.
> > > > > > >
> > > > > > > In summary, CPQ lowers the Q-value by omission (dropping the future), which may be insufficient if the immediate reward is high. CARL lowers the Q-value by direct penalization (replacing the reward), which guarantees that the state-action pair is assigned a low value regardless of the immediate reward magnitude.
> > > > > > >
> > > > > > > ---
> > > > > > >
> > > > > > > ## Q6
> > > > > > >
> > > > > > > ### 1. The Update Formulas for IQL-based CARL
> > > > > > >
> > > > > > > The reviewer asks for the explicit formulation to verify the update style. In CARL, the Cost Critic ($Q_c$) and the Reward Critic ($Q_r$) are updated as follows:
> > > > > > >
> > > > > > > **Cost Critic ($Q_c$ - AC Style):**
> > > > > > > Updated via standard Bellman Expectation with the current policy $\pi$:
> > > > > > > $$
> > > > > > > L_{Q_c}(\theta) = \mathbb{E}_{(s,a,s',c) \sim \mathcal{D}} \left[ (c + \gamma Q_c(s', \pi(s')) - Q_c(s,a))^2 \right]
> > > > > > > $$
> > > > > > >
> > > > > > > **Data Relabeling (The Interface):**
> > > > > > > For a transition $(s,a)$ in the batch:
> > > > > > > $$
> > > > > > > r_{\text{new}}(s,a) = \begin{cases}
> > > > > > > -V_{max} & \text{if } Q_c(s,a) > \kappa,  \\\\
> > > > > > > r(s,a) & \text{otherwise}
> > > > > > > \end{cases}
> > > > > > > $$
> > > > > > >
> > > > > > > **Reward Critic ($Q_r$ - IQL Style):**
> > > > > > > Updated via Expectile Regression using strictly dataset actions (no policy query):
> > > > > > >
> > > > > > > $$
> > > > > > > L_{Q\_r}(\phi) = \mathbb{E}\_{(s,a,s') \sim \mathcal{D}} \big[ L\_2^\tau( r\_{\mathrm{new}}(s,a) + \gamma V(s') - Q(s,a) ) \big]
> > > > > > > $$
> > > > > > >
> > > > > > >
> > > > > > >
> > > > > > > (Where $V(s')$ is the IQL value network trained via expectiles).

---

> > > > > > > > ### Author Response · Authors · 2025-12-04
> > > > > > > >
> > > > > > > > ### 2. Why AC-style $Q_c$ does not break IQL properties
> > > > > > > >
> > > > > > > > $Q_c$ is updated in an "AC-style," which potentially introduces OOD errors into the cost estimates. However, we respectfully disagree with the conclusion that this makes CARL equivalent to TD3+BC or that it breaks the IQL backbone.
> > > > > > > >
> > > > > > > > * **IQL Integrity ($Q_r$):** The core logic of IQL comes from never bootstrapping on OOD actions for the reward value function. In CARL, $Q_r$ is trained on the static dataset $\mathcal{D}$ with relabeled rewards $r_{\text{new}}$. Even if $Q_c$ has OOD errors, these manifest only as label noise in $r_{\text{new}}$ – we already demonstrated that CARL is robust to relabeling noise in our ablation experiments in the rebuttal. The $Q_r$ update mechanism remains strictly IQL (expectile-based). It never queries the policy $\pi$ for bootstrapping.
> > > > > > > > * **Cost Critic Role:** We utilize $Q_c$ only to evaluate transitions $(s,a)$ that actually exist in the dataset (to decide on relabeling). We do not use $Q_c$ to bootstrap the reward critic.
> > > > > > > > * **Robustness to $Q_c$ Error:** The reviewer implies that if $Q_c$ has OOD errors, the system fails. However, our noise-robustness ablations in the rebuttal demonstrate that CARL is highly resilient to noisy safety signals and does not require high-precision $Q_c$ value estimation.
> > > > > > > >
> > > > > > > > ### 3. Comparison with C2IQL
> > > > > > > >
> > > > > > > > The reviewer mentions C2IQL to highlight the necessity of "rigid ties." C2IQL couples the cost and reward expectiles because it attempts to learn both implicitly.
> > > > > > > >
> > > > > > > > CARL takes a different approach: we accept the explicit (AC-style) learning of costs to maintain modularity. This allows CARL to act as a wrapper. CARL empirically shows that one can combine an **Explicit Safety Check** ($Q_c$) with an **Implicit Value Learner** ($Q_r$) effectively to learn safe and reward-maximizing policies.

---

### Official Review · Reviewer_chYS · 2025-11-01

**Soundness:** 3
**Presentation:** 3
**Contribution:** 3
**Rating:** 4
**Confidence:** 3

**Summary:**

This paper presents Constraint-Aware Reward Relabeling (CARL), a straightforward and hyperparameter-free approach to offline safe reinforcement learning (OSRL). CARL enforces safety constraints without using Lagrange multipliers or dual optimization by alternating between cost estimation and policy improvement. During training, rewards for actions predicted to be unsafe are replaced with strong negative penalties, effectively turning the constrained objective into an unconstrained one. The method serves as a lightweight wrapper that can be integrated with existing offline RL algorithms such as TD3-BC and IQL, enabling the learning of safe yet high-performing policies even from datasets containing many unsafe samples. Experiments on the DSRL benchmark show that CARL achieves consistently strong results, meeting cost constraints while maintaining high returns across various tasks and cost settings, highlighting its stability, generality, and practical utility for safety-critical offline RL.

**Strengths:**

- CARL provides a clean and practical approach to offline safe RL, removing the need for complex constrained optimization yet achieving strong safety and reward trade-offs. The method’s simplicity and consistent empirical gains make it a notable contribution.
- Because CARL operates solely through reward relabeling at the data-processing stage, it can be seamlessly paired with diverse offline RL methods, making it a flexible and practical choice for safety-critical deployment.
- The paper offers a clear theoretical contribution by showing that optimizing a relabeled reward function with large penalties for unsafe actions leads to policies that satisfy pointwise safety constraints, and is formally equivalent to the original constrained CMDP formulation under mild assumptions.
- The paper is well written, with clear organization and logical flow throughout. In particular, the results are effectively visualized—figures and tables are well designed, easy to interpret, and enhance the overall clarity and impact of the presentation.

**Weaknesses:**

- In the CARL framework, performance is highly dependent on the accuracy of the cost evaluation function. Inaccurate cost-to-go estimates can result in incorrect reward relabeling, which introduces significant uncertainty and may compromise both safety and reward performance. However, the paper does not include any sensitivity analysis to assess how estimation errors in the cost value impact the overall behavior of the learned policy, leaving an important aspect of robustness unaddressed.
- Although the paper reformulates the OSRL objective as an unconstrained optimization problem, it lacks a theoretical convergence analysis of the proposed iterative procedure, especially in the context of value function approximation. Providing such an analysis, or clearly outlining conditions under which convergence is guaranteed, would significantly improve the theoretical soundness and credibility of the method.
- This paper claims that CARL can combine with any other offline RL algorithm framework, but only two frameworks were tested — both are value-based, not generative or transformer-based (e.g., CQL, Diffuser, Decision Transformer) [1-3].
- This paper lacks comparisons with several recent and strong baselines, which makes it difficult to fully assess the claimed performance improvements and the novelty of the proposed method relative to the current state of the art [4-6].
- While CARL is designed as a minimalist wrapper, it introduces an additional value function training phase for cost estimation in each iteration. This added computational step—particularly when used with deep offline RL backbones—may impact training time and resource usage. However, the paper does not provide any analysis or empirical comparison of runtime efficiency, training wall-clock time, or computational complexity relative to baseline methods [7].
- As shown in Table 1, CARL occasionally fails to satisfy the cost constraint in certain tasks, and in some settings, it does not achieve the highest reward compared to other baselines. The paper would benefit from a deeper analysis of these cases to understand the conditions under which CARL underperforms, and to clarify the trade-offs between safety and performance [8].

**Questions:**

- How sensitive is CARL’s performance to errors in the cost-to-go estimator Qc?
- Are there theoretical guarantees that CARL will respect constraints under bounded Qc​ estimation error?
- Could you evaluate CARL’s compatibility with non–value-based offline RL frameworks such as CQL, AWAC, or Decision Transformer?
- Could you provide a performance comparison between CARL and additional recent baselines, including those in references [4–6]?

## Reference

**[1]** Janner, Michael, et al. "Planning with diffusion for flexible behavior synthesis." arXiv preprint arXiv:2205.09991 (2022).

**[2]** Chen, Lili, et al. "Decision transformer: Reinforcement learning via sequence modeling." Advances in neural information processing systems 34 (2021): 15084-15097.

**[3]** Kumar, Aviral, et al. "Conservative q-learning for offline reinforcement learning." Advances in neural information processing systems 33 (2020): 1179-1191.

**[4]** Wang, Ruhan, and Dongruo Zhou. "Safe Decision Transformer with Learning-based Constraints." 7th Annual Learning for Dynamics\& Control Conference. PMLR, 2025.

**[5]** Guan, Jiayi, et al. "Voce: Variational optimization with conservative estimation for offline safe reinforcement learning." Advances in Neural Information Processing Systems 36 (2023): 33758-33780.

**[6]** Wei, Honghao, et al. "Adversarially trained weighted actor-critic for safe offline reinforcement learning." Advances in Neural Information Processing Systems 37 (2024): 52806-52835.

**[7]** Chemingui, Yassine, et al. "Constraint-adaptive policy switching for offline safe reinforcement learning." Proceedings of the AAAI Conference on Artificial Intelligence. Vol. 39. No. 15. 2025.

**[8]** Gong, Ze, Akshat Kumar, and Pradeep Varakantham. "Offline safe reinforcement learning using trajectory classification." Proceedings of the AAAI Conference on Artificial Intelligence. Vol. 39. No. 16. 2025.

---

> ### Author Response · Authors · 2025-11-21
>
> We thank the reviewer for their feedback and for taking the time to evaluate our work. We address their comments below:
> ## **Theoretical convergence analysis:**
>
> While in general it is possible to design pathological MDPs that result in non-convergence, it seems highly plausible that there are reasonable conditions for a class of MDPs where convergence could be guaranteed.
>
> However, we argue that the lack of a theoretical guarantee is counterbalanced by the method's specific design goal: to empirically stabilize the oscillation observed in Lagrangian methods.
>    - Design Motivation: As detailed in Section 5, standard iterative solvers often lead to severe oscillations where the policy and cost critic diverge. Our $M=K=1$ schedule is designed to force these components to track each other closely, acting as a dampening mechanism.
>    - Empirical Validation: While we lack a formal proof, our extensive empirical evaluation across 19 tasks demonstrates that this procedure reliably learns  safe solutions in practice, outperforming theoretically strong  but practically brittle baselines.
>
>
> ## **$Q_c$ accuracy:**
>
> We address the core concern, robustness to estimation error for Q_c, through two targeted stress tests.
>   - **Experiment A: Noise Injection:** During training, we randomly invert the decision of the cost critic (flipping "safe" to "unsafe" and vice versa) with a probability `p ∈ {10%, 20%, 30%, 50%}` to inject noise.
>
> Result: Our results demonstrate that the policy remains safe under moderate levels of noise. Using normalized cost < 1 as the safety criterion, all evaluated tasks remain safe at noise levels of 10% and 20%, despite mild degradation in rewards. This indicates that performance deteriorates before safety is compromised, which is the intended behavior of our safety mechanism. At 30% noise, BallCircle becomes unsafe (normalized cost 2.85) and DroneCircle gets negative reward −0.15, while Carcircle and AntCircle remain safe. Finally, at 50% noise, all tasks incur large safety violations (normalized costs 10–20), consistent with behavior driven by near-random inputs. Importantly, this regime corresponds to noise levels where the input signal is largely uninformative. The fact that safety fails only when relabeling decisions become effectively random supports our claim that the CARL method is safe under plausible perturbations but cannot be expected to remain safe under adversarially extreme noise levels.
>
> | Noise | Task        | Reward            | Cost              |
> |-------|-------------|------------------|------------------|
> | **0.1** | BallCircle  | 0.73 ± 0.01       | 0.35 ± 0.24       |
> |       | CarCircle   | 0.70 ± 0.01       | 0.02 ± 0.03       |
> |       | DroneCircle | 0.52 ± 0.03       | 0.00 ± 0.00       |
> |       | AntCircle   | 0.60 ± 0.02       | 0.09 ± 0.10       |
> | **0.2** | BallCircle  | 0.75 ± 0.01       | 0.11 ± 0.05       |
> |       | CarCircle   | 0.69 ± 0.04       | 0.00 ± 0.00       |
> |       | DroneCircle | 0.55 ± 0.01       | 0.00 ± 0.00       |
> |       | AntCircle   | 0.47 ± 0.05       | 0.00 ± 0.00       |
> | **0.3** | BallCircle  | 0.49 ± 0.33       | 2.85 ± 4.61       |
> |       | CarCircle   | 0.71 ± 0.03       | 0.00 ± 0.00       |
> |       | DroneCircle | -0.15 ± 0.12      | 0.48 ± 0.44       |
> |       | AntCircle   | 0.33 ± 0.19       | 0.71 ± 0.27       |
> | **0.5** | BallCircle  | 0.87 ± 0.02       | 10.56 ± 0.81      |
> |       | CarCircle   | 0.90 ± 0.01       | 16.73 ± 0.44      |
> |       | DroneCircle | 0.50 ± 0.53       | 13.40 ± 7.13      |
> |       | AntCircle   | 0.70 ± 0.08       | 20.62 ± 3.34      |
>
>
> This experiment directly addresses the reviewer's concern regarding sensitivity to OPE miscalibration.
>
> The results demonstrate that CARL does not rely on precise, pointwise accuracy of the cost estimator ($Q_c$). Instead, the method is functionally robust to significant signal degradation. Even when nearly one-third of the safety labels are noisy, the aggregate topological signal provided by the critic remains strong enough to guide the policy into the feasible region.
> This resilience validates the design choice of the iterative update cycle ($M=K=1$). By co-evolving the policy and critic incrementally, transient estimation errors are "washed out" over the course of training rather than accumulating catastrophically.
> The collapse of safety at the 50% noise level serves as a critical sanity check. It confirms that the safety observed in standard runs is causally driven by the $Q_c$ signal, and not by external factors or environment design,  demonstrating that the critic is indeed providing the necessary directional guidance.

---

> > ### Author Response · Authors · 2025-11-21
> >
> > - **Experiment B: Random Penalties:** we removed the $Q_c$ signal entirely and instead penalized a random subset of transitions (e.g., 70-90%).
> >
> > Result: The agent failed to achieve safety across the tasks.
> >
> > | Percentage | Task        | Reward         | Cost            |
> > |------------|-------------|----------------|-----------------|
> > | **0.7**    | BallCircle  | 0.54 ± 0.36     | 6.86 ± 1.06      |
> > |  | CarCircle   | 0.84 ± 0.03     | 15.09 ± 0.35     |
> > |            | DroneCircle | 0.44 ± 0.52     | 8.68 ± 6.11      |
> > |            | AntCircle   | 0.64 ± 0.02     | 18.83 ± 2.54     |
> > | **0.9**    | BallCircle  | 0.01 ± 0.01     | 11.84 ± 20.50    |
> > |   | CarCircle   | 0.76 ± 0.03 | 13.44 ± 1.33  |
> > |  | DroneCircle | 0.49 ± 0.53  | 9.49 ± 6.83  |
> > |   | AntCircle   | 0.60 ± 0.07  | 17.47 ± 1.50 |
> >
> > This experiment refutes the potential critique that CARL achieves safety merely through over-conservatism or indiscriminate data suppression.
> > The failure of the random variant demonstrates that the information content of the $Q_c$ signal is necessary. Safety in these environments is not achieved simply by avoiding a large percentage of the state-action space, but by avoiding specific, structurally unsafe transitions.
> > This  corroborates our hypothesis that despite the inherent difficulty of OPE, the cost critic is successfully identifying the correct manifold of unsafe state-action pairs. The penalties in CARL are not just regularization; they are effectively shaping the decision boundary. Random penalties disrupt this boundary, leading to constraint violations, whereas $Q_c$-guided penalties (even with the noise demonstrated in Experiment A) successfully preserve it.
> >
> >
> > ## **Non Value-based backbone:**
> > We appreciate the reviewer pushing for broader architectural validation. We agree that verifying CARL on a fundamentally different paradigm, such as sequence modeling, strengthens the generality claim. To address this, we conducted a new experiment applying CARL to a Decision Transformer (DT) backbone based on the CDT implementation, without conditioning on costs. Since DT operates via supervised learning on trajectory histories rather than iterative Bellman updates, we adapted the workflow as follows:
> > - Cost Estimation: We trained a cost critic ($Q_c$) on the behavior dataset for 5k update steps.
> > - Relabeling: We utilized this learned critic to relabel the dataset rewards according to the CARL mechanism (Eq. 5), assigning penalties to state-action pairs where $Q_c(s,a) > \kappa$.
> > - Sequence Modeling: We then trained a standard Decision Transformer on this relabeled data for 20k update steps.
> >
> > Results: We evaluate CARL-DT on six tasks under a cost limit of 5, following the same protocol as the main experiments. Despite the reduced training budget (CDT typically use 100k steps), the CARL-DT agent successfully satisfied safety constraints. This confirms that the reward relabeling principle of CARL is effective not just for value-based actor-critic methods, but also for sequence-modeling architectures that learn purely via supervised objectives.
> > | Task| reward | cost |
> > |-|-|-|
> > | BallRun| 0.22 ± 0.07 | 0.00 ± 0.00 |
> > | CarRun| 0.93 ± 0.06 | 0.00 ± 0.00 |
> > | DroneRun| 0.54 ± 0.01 | 0.00 ± 0.00 |
> > | BallCircle| 0.38 ± 0.07 | 0.22 ± 0.27 |
> > | CarCircle| 0.47 ± 0.04 | 0.49 ± 0.62 |
> > | DroneCircle| 0.48 ± 0.04 | 0.49 ± 0.59 |
> >
> > ## **More Baselines Comparison:**
> > We prioritized comparisons against established, code-accessible methods that represent the main families of OSRL (Lagrangian, Diffusion/Feasibility, Trajectory Transformer). Regarding the specific papers mentioned:
> > - Baseline [4] (DT-based): This method shares significant architectural similarity with the Constrained Decision Transformer (CDT), which we already include as a primary baseline. Furthermore, the official code for [4] is not available.
> > - Baseline [5] (VOCE) & [6] (WSAC): We conducted preliminary evaluations of these methods using their default configurations on the tight-budget regime ($\kappa=5$) used in our benchmarks, following the same protocol as the main experiments. We found that they failed to satisfy the safety constraints in this challenging setting. Specifically, WSAC [6] is primarily a theoretical contribution and was only empirically evaluated on a small set of four tasks, all under a relatively relaxed cost threshold ($\kappa = 40$). We ran their implementation with the provided parameters without additional adaptation or tuning; under these conditions, it did not satisfy the strict safety budget of our benchmark ($\kappa = 5$), except for AntRun where rewards were almost 0. VOCE also failed to meet safety constraints in this setting, except for AntRun and AntCircle, which achieved very low rewards. We did not tune VOCE beyond its default hyperparameters. Additionally, because VOCE was trained on its own collected dataset rather than the data used in our experiments, differences in data distribution may further contribute to the performance gap.

---

> > > ### Author Response · Authors · 2025-11-21
> > >
> > > Since a core principle of CARL is its "minimalist" nature, working across tasks without hyperparameter tuning, we believe the comparison against the most recent and strongest performing baselines (FISOR, CAPS, CCAC) adequately establishes its state-of-the-art status.
> > >
> > > WSAC results:
> > > | Task | reward | cost |
> > > |------------------------|--------|------|
> > > | BallRun               | 0.37 ± 1.46 | 18.40 ± 0.35 |
> > > | CarRun                | 1.06 ± 0.19 | 25.28 ± 10.39 |
> > > | DroneRun              | 0.57 ± 0.59 | 24.47 ± 6.49 |
> > > | AntRun                | 0.01 ± 0.01 | 0.00 ± 0.00 |
> > > | BallCircle            | 0.28 ± 0.48 | 16.42 ± 19.11 |
> > > | CarCircle             | 0.56 ± 0.10 | 17.05 ± 3.39 |
> > > | DroneCircle           | -0.26 ± 0.00 | 2.46 ± 4.04 |
> > > | AntCircle             | 0.03 ± 0.05 | 4.15 ± 7.18 |
> > >
> > > Voce Results:
> > > | Tasks | reward | cost |
> > > |--|--|--|
> > > | BallRun | 0.43 ± 0.09 | 4.07 ± 1.66 |
> > > | CarRun | 0.78 ± 0.29 | 6.65 ± 3.88 |
> > > | DroneRun | 0.07 ± 0.07 | 4.68 ± 3.76 |
> > > | AntRun | 0.09 ± 0.02 | 0.02 ± 0.02 |
> > > | BallCircle | 0.61 ± 0.08 | 1.56 ± 0.47 |
> > > | CarCircle | 0.33 ± 0.16 | 5.14 ± 0.22 |
> > > | DroneCircle | -0.27 ± 0.00 | 7.24 ± 1.32 |
> > > | AntCircle | 0.00 ± 0.00 | 0.00 ± 0.00 |
> > >
> > > ## **CARL occasionally fails:**
> > > The reviewer correctly notes that CARL is not a "silver bullet" and fails on a small subset of tasks (3 out of 19). We offer the following analysis for these trade-offs:
> > >
> > > - Consistency vs. Peak Performance: No single method solves every OSRL task perfectly. However, CARL achieves a significantly higher success rate than the state-of-the-art. As shown in Table 1, CARL satisfies safety constraints on 16 out of 19 tasks. In comparison, the next-best methods (FISOR and CAPS) are safe on only roughly half the tasks (10 out of 19), and methods  such as CPQ are safe on even fewer.
> > >
> > > - The "Minimalist" Trade-off: The failures likely stem from our strict "no-tuning" policy. We use a fixed penalty formulation ($R_{max}$) and fixed update schedule ($M=K=1$) across all environments to demonstrate generalization. It is likely that task-specific tuning would resolve these specific failures. However, we deliberately refrain from such tuning to present an honest view of the method's out-of-the-box robustness.
> > > The trade-off presented by CARL is generalization over specialization. We accept rare failures in exchange for a method that requires no hyperparameter sweeping and works reliably on the vast majority of tasks.
> > >
> > > ## **Runtime Comparison**
> > >
> > > When running without any competing workloads, training a single task for one random seed takes approximately 31 minutes per 200k update steps. Running TD3BC without Qc is around 21 minutes.

---

### Official Review · Reviewer_LTcd · 2025-11-03

**Soundness:** 2
**Presentation:** 2
**Contribution:** 2
**Rating:** 4
**Confidence:** 4

**Summary:**

The paper proposes **Constraint-aware Reward (Re)Labeling (CARL)** for offline safe RL. It learns a cost-to-go critic and relabels rewards in each mini-batch: if $Q_c^\pi(s,a)>\kappa$, assign a large negative penalty; otherwise keep the original reward. An off-the-shelf offline RL backbone (e.g., TD3-BC, IQL) is then trained on the relabeled data.

**Strengths:**

* **Practical motivation:** avoids brittle dual updates seen in some Lagrangian methods.
* **Intuitive method:** convert safety constraints into state-action pair penalties rather than tuning Lagrange multipliers.
* **Simple backbone-agnostic wrapper:** CARL can be wrapped around standard offline RL backbones.

**Weaknesses:**

## Positioning & Objectives
* **Tight-budget regime but CMDP formulation:** The paper targets small cost limit $\kappa$ and *pointwise safety*, which is closer to *hard-constraint / shielded* or *risk-sensitive* formulations. However, problem setup is based on CMDP and baselines are CMDP-style OSRL only.
* **Theory-metric mismatch:** Theory enforces *statewise* safety; but evaluation reports *episodic normalized cost*. If pointwise safety is the goal, episodic evaluation metric may be a mismatch.
* **Feasibility under offline coverage:** Pointwise constraints may be *infeasible* for a given $\kappa$ with partial data coverage while the expectation constraints are *feasible*. Current submission does not discuss this scenario in detail.

## Method & Claims
* **Theorem 1 conditions (notation/assumption):** The proof sketch relies on a sufficiently punitive $-V_{\max}$, feasibility of the pointwise-constrained problem, and bounded nonnegative (or shifted-to-nonnegative) rewards. Note that nonnegative reward is not specified in problem setup, but it is required for $V_r^{\tilde{\pi}*}(s) > 0$ used in the proof of Theorem 1. In addition, the main experiments use $-R_{\max}$, not $-V_{\max}$, weakening the applicability of Theorem 1.
* **CARL stability vs Lagrangian:** CARL replaces dual updates with reward relabeling using Q-estimates. However, this introduces non-stationarity between the policy, fitted Q evaluation (FQE), and relabeling; and inherits FQE’s estimation noise. With the absence of convergence analysis, it’s unclear that CARL is less brittle than Lagrangian methods. Rather, it may swap dual-instability for OPE-instability.
* **OPE dependence:** Unsafe detection hinges on FQE accuracy under distribution shift; miscalibration can over- or under-penalize.
* **“Neighborhood suppression” claim:** With function approximation, penalizing one transition can generalize unpredictably beyond a "local neighborhood".
* **"No extra hyperparameters" claim:** The value of *penalty magnitude* has to be determined. The paper uses ($R_{\max}$ vs $V_{\max}$) for experiments. However, this value is offline dataset dependent and there may not be a sufficiently good estimate at deployment time. In addition, this value varies significantly across tasks.

## Evaluation
* **Baselines:** For tight budget setting, **hard-constraint/shielded** and **risk-sensitive** baselines may be more appropriate for small $\kappa$.
* **Budgets & metrics:** If the aim is strict/tight safety, violation rate and/or per-state safety metrics should be reported alongside episodic cost.
* **Statistical confidence:** Evaluation results average over 20 episodes and 3 random seeds. This seems too few to establish statistical confidence.

## Reproducibility
* The anonymous code link given in the paper is broken. No code is accessible, the link returns "The requested file is not found."

## Minor Typo
* **Line 088:** “ation-value”.

**Questions:**

## Theory & Guarantees
1. The paper uses $-R_{\max}$ as the penalty magnitude; what guarantee remains in practice, since Theorem 1 depends on $-V_{\max}$?

## Method & Stability
2. How is the **penalty scale** chosen at deployment (task-agnostic guidance)?
3. How much noise do you witness in the OPE relative to threshold $\kappa$? How often are actions misclassified as unsafe/safe?
4. Does CARL update exhibit contraction/monotone improvement? My assessment is that the paper should temper the claim that CARL alleviates Lagrangian brittleness. It is not a strictly better stability trade-off.

---

> ### Author Response · Authors · 2025-11-21
>
> We thank the reviewer for their feedback and for taking the time to evaluate our work. We address their comments below:
>
> - **Tight-budget regime but CMDP formulation:**
>
> We appreciate the reviewer raising this nuance. We clarify that we do not propose a new problem definition; our goal remains solving the standard CMDP problem defined by episodic cost constraints (Eq. 1).  We utilize the stricter, pointwise safety formulation (Eq. 2) specifically as a methodological tool, a sufficient condition, to solve the original CMDP problem. As noted in Theorem 1, a solution to the pointwise problem immediately yields a solution to the CMDP problem. Our motivation for this formulation/approach is practical rather than theoretical. While Lagrangian methods are principled, they can be difficult to tune in the offline setting, where gradient updates on the multipliers ($\lambda$) often lead to instability or oscillation due to extrapolation errors in value estimation. Indeed, our experience and difficulties in getting Lagrangian methods to work in practice motivated this alternative approach. By converting the problem into an unconstrained objective (Eq. 3), we bypass the specific challenge of tuning Lagrangian multipliers while still targeting the original CMDP goal.
>
> We included FISOR as a state-of-the-art baseline to represent the class of hard-constraint/feasibility-based methods for CMDPs. While these methods are relevant, our problem formulation follows the standard Constrained MDP (CMDP) setup defined in Equation 1. Therefore, it is necessary to compare against the standard suite of CMDP-based OSRL algorithms (CPQ, COptiDICE, CCAC etc.) to demonstrate that CARL advances the state-of-the-art for this specific problem definition.
> - **Theory-metric mismatch:**
>
> We agree with the reviewer that our theory leverages state-wise safety, while our evaluation uses episodic cost. This is intentional and consistent with our  problem definition described above. Since the pointwise formulation (Eq. 2) is a sufficient condition for the episodic CMDP formulation (Eq. 1), maximizing the former theoretically guarantees satisfaction of the latter. Therefore, evaluating on the standard episodic normalized cost metric is the correct way to verify if our "sufficient condition" strategy successfully solves the original OSRL problem.
>
> We follow the evaluation protocol of the DSRL benchmark  to ensure fair comparison. The results confirm that our state-wise wrapper approach successfully translates into superior episodic safety performance, validating that solving the stricter proxy problem is an effective strategy for the standard episodic task.
>
> - **Feasibility under offline coverage:**
>
> The reviewer makes a good point: pointwise constraints define a smaller feasible region than expectation constraints, which could be problematic under partial dataset coverage.  The set of policies satisfying the pointwise constraint (Eq. 2) is indeed a subset of those satisfying the episodic constraint (Eq. 1). Therefore, with limited offline data coverage, it is possible that a solution exists for the episodic problem but not for the stricter pointwise problem.
>
> We acknowledge this as a design trade-off: we prioritize a formulation that enables a stable, unconstrained solution approach (Eq. 3) over maximizing the theoretical feasible set. In practice, however, the pointwise constraint is far less brittle than the theoretical comparison suggests because it is enforced only on the reachable state–action pairs induced by the learned policy, not the entire offline dataset support. Thus feasibility depends primarily on local coverage near the policy’s trajectory distribution.
> We provide empirical evidence that this stricter feasible set remains accessible even under data limitations. In our ablation study using only unsafe trajectories (Table 5), the behavior policy $\pi_\beta$ covers a region of the state-action space that predominantly yields high costs. Even in this scenario, where no safe trajectories exist in the dataset, CARL successfully synthesizes a policy that satisfies the constraint!
>
> - **Theorem 1 conditions (notation/assumption):**
>
> We address the gap between the theoretical bound and the practical implementation as follows:
> $R_{max}$ vs. $V_{max}$: We explicitly analyze this choice in Appendix B.1. While $-V_{max}$ provides the theoretical guarantee for the rigorous pointwise constraint, it is overly conservative in practice, leading to lower rewards. We opted for $-R_{max}$ because it offers a superior empirical trade-off between safety and reward without requiring sensitivity analysis to the discount factor $\gamma$.
>
> Non-negative Rewards: The assumption $V_r > 0$ is required for the proof sketch but is mostly formal. For the DSRL benchmark tasks, returns are effectively non-negative (Fig 3).
>
> In summary, the theorem motivates the reward-relabeling structure, while the penalty magnitude is chosen empirically to balance safety and reward.

---

> > ### Author Response · Authors · 2025-11-21
> >
> > - **CARL stability vs Lagrangian:**
> >
> > The reviewer raises a fair point: replacing dual updates with OPE-dependent relabeling swaps one source of potential instability for another. We do not claim to solve the non-stationarity problem; rather, we argue that our trade-off is more favorable for offline learning.
> >
> > Empirically, our results demonstrate that this theoretical concern does not translate into practical brittleness. Across the 19 benchmark tasks, CARL successfully produces safe policies on 16 of them. In contrast, the strongest baselines (FISOR and CAPS) are safe on only 10 tasks, while Lagrangian-based methods fare significantly worse: COptiDICE and BEAR-Lag are safe on only 2 tasks, and BCQ-Lag fails to satisfy constraints only on one task. We include the results for BEAR-Lag and BCQ-Lag, which are Lagrangian extensions of the offline RL methods BEAR and BCQ below. These variants have been previously reported to fail in the original DSRL paper, as well as in later OSRL works. We observe similar behavior under our evaluation settings.
> >
> > | Method   | | BallRun | CarRun | DroneRun | AntRun | BallCircle | CarCircle | DroneCircle | AntCircle | CarCircle1 |
> > |----------|--------|---------|--------|----------|--------|------------|-----------|-------------|-----------|------------|
> > | BCQ-Lag  | Reward | 0.67 ± 0.24 | 0.86 ± 0.08 | 0.74 ± 0.06 | 0.81 ± 0.05 | 0.69 ± 0.03 | 0.62 ± 0.05 | 0.66 ± 0.05 | 0.54 ± 0.10 | 0.66 ± 0.04 |
> > |   | Cost   | 11.33 ± 3.69 | 0.83 ± 0.76 | 21.45 ± 2.50 | 22.09 ± 2.54 | 7.55 ± 1.95 | 8.73 ± 1.78 | 6.09 ± 3.08 | 9.39 ± 3.21 | 15.83 ± 2.76 |
> > | BEAR-Lag | Reward | 0.53 ± 0.47 | 0.27 ± 1.20 | -0.18 ± 0.09 | 0.01 ± 0.02 | 0.85 ± 0.04 | 0.74 ± 0.06 | 0.86 ± 0.07 | 0.69 ± 0.03 | 0.81 ± 0.02 |
> > |  | Cost   | 18.60 ± 0.35 | 30.27 ± 10.48 | 16.31 ± 9.49 | 0.00 ± 0.01 | 10.54 ± 1.74 | 7.51 ± 1.45 | 15.90 ± 2.51 | 19.03 ± 0.84 | 19.07 ± 1.09 |
> >
> > | Method   | | CarCircle2 | CarGoal1 | CarGoal2 | PointCircle1 | PointCircle2 | PointGoal1 | PointGoal2 | AntVelo | HalfChehVelo | SwmrVelo |
> > |----------|--------|------------|----------|----------|---------------|---------------|------------|------------|-------------|----------------------|------------------|
> > | BCQ-Lag  | Reward | 0.62 ± 0.08 | 0.47 ± 0.07 | 0.30 ± 0.03 | 0.56 ± 0.15 | 0.44 ± 0.07 | 0.73 ± 0.01 | 0.68 ± 0.02 | 1.00 ± 0.01 | 1.05 ± 0.01 | 0.51 ± 0.08 |
> > |   | Cost   | 18.21 ± 2.36 | 2.41 ± 1.14 | 4.35 ± 0.51 | 8.28 ± 5.58 | 2.98 ± 1.39 | 4.03 ± 0.22 | 9.85 ± 1.51 | 7.11 ± 3.65 | 61.93 ± 5.33 | 25.01 ± 3.21 |
> > | BEAR-Lag | Reward | 24.73 ± 3.06 | 3.63 ± 0.83 | 8.13 ± 1.25 | 15.29 ± 8.52 | 23.50 ± 6.85 | 3.69 ± 0.93 | 11.92 ± 1.17 | 0.00 ± 0.00 | 5.72 ± 3.12 | 9.33 ± 5.06 |
> > |  | Cost   | 24.73 ± 3.06 | 3.63 ± 0.83 | 8.13 ± 1.25 | 15.29 ± 8.52 | 23.50 ± 6.85 | 3.69 ± 0.93 | 11.92 ± 1.17 | 0.00 ± 0.00 | 5.72 ± 3.12 | 9.33 ± 5.06 |
> >
> >
> > This stark difference highlights that while Lagrangian methods possess theoretical convergence guarantees, they are difficult to tune in the offline setting, often resulting in brittleness and constraint violation in practice. CARL’s high success rate suggests it effectively stabilizes the learning process where other methods fail, even  when they are theoretically-sound.
> > Furthermore, to confirm that CARL is not simply "lucky" with its OPE estimates, we conducted a robustness ablation where we randomly flipped the $Q_c$ based safety decision (a given transition from offline data is safe or not) with varying probabilities. We found that even with significant label noise, CARL consistently learned safe policies, indicating that the method is robust to the OPE estimation errors  which might otherwise destabilize a Lagrangian approach. This ablation details are available in the response to reviewer chYS.
> >
> > - **OPE dependence:**
> >
> > We agree that OPE accuracy is a critical bottleneck. However, this challenge is universal for all methods which require cost/reward estimation: Lagrangian baselines and methods such as  CAPS also fundamentally depend on accurate cost estimation ($Q_c$) to function. (Only methods such as CDT or BC-Safe avoid this specific dependency).
> >
> > The key difference lies in the mechanism of error propagation. Lagrangian methods update the multiplier $\lambda$ based on the magnitude of the constraint violation; therefore, errors in $Q_c$ can drive $\lambda$ arbitrarily, causing the policy to fail. CARL employs a binary mechanism: if the cost limit is violated, a fixed penalty is applied. This structure prevents estimation errors from scaling the penalty magnitude.
> >
> > Our robustness ablation (randomly flipping $Q_c$ decisions for safe/unsafe transitions) confirms that this mechanism remains effective even under significant noise. While OPE errors may indeed lead to occasional over- or under-penalization in individual steps, the aggregate empirical results (Table 1) demonstrate that CARL consistently satisfies constraints where baselines fail, achieving a superior practical trade-off between safety and reward.

---

> > > ### Author Response · Authors · 2025-11-21
> > >
> > > - **“Neighborhood suppression” claim:**
> > >
> > > The reviewer correctly notes that function approximation can generalize penalties unpredictably beyond a local neighborhood. We address this risk through two specific design choices:
> > >
> > > Iterative Stability ($M=K=1$): As described in Section 5, we restrict updates to a single step ($M=K=1$) to minimize the divergence between the policy and the cost critic. This forces the cost estimates to track the policy update incrementally, preventing large shifts in the cost-to-go function that would otherwise lead to broad, inaccurate penalty generalization across the state space.
> > >
> > > Backbone Constraints: CARL utilizes offline RL backbones that explicitly constrain the policy to the behavior distribution. For example, our implementation uses TD3-BC, which applies a behavior cloning regularizer, and IQL, which learns value functions strictly from in-sample actions. These mechanisms prevent the policy from querying the value function in out-of-distribution regions, where approximation error and unpredictable generalization are most severe.
> > >
> > > Furthermore, our robustness ablation (randomly flipping $Q_c$ based decisions for safe/unsafe transitions) provides empirical evidence that precise penalty boundaries are not required for success. The results indicate that the method remains effective even when the penalty signal is noisy, suggesting that the learned policy is robust to the specific approximation errors introduced by the relabeling process.
> > >
> > > - **"No extra hyperparameters" claim:**
> > >
> > > We agree with the reviewer that the penalty magnitude is technically a design choice. However, we distinguish between a tunable hyperparameter (which requires grid search) and a dataset-derived statistic.
> > > We define "minimalist" in the context of avoiding the need for task-specific tuning. The penalty in CARL is calculated automatically from the offline dataset ($R_{max}$ or $V_{max}$).  CARL doesn’t require any type of estimation at deployment time (as the policy is fixed after training).
> > >
> > > - **Budgets & metrics:**
> > >
> > > We adhere to the standard DSRL evaluation protocol, which defines success based on the episodic cost constraint (Eq. 1) rather than per-state safety. The pointwise constraint (Eq. 2) is utilized strictly as a training mechanism to achieve this episodic goal. However, we agree that violation rate is a valuable auxiliary metric. We include the episode violation rate (percentage of test trajectories that satisfy the cost constraint) to provide a more granular view of safety performance, in our results with extra seeds below.
> > >
> > > - **Statistical confidence:**
> > >
> > > This is the regular setup for  evaluation in DSRL benchmarks and is the same one used by prior OSRL methods.
> > > We run CARL for three additional seeds (original seeds: 10, 20, 30; added seeds: 40, 50, 60) and report results following the same protocol as the main experiments. We also report both the count and percentage of safe trajectories. Each task includes 120 total trajectories.
> > > | Task     | reward         | cost          | N safe | % safe |
> > > |------------------|----------------|---------------|--------|--------|
> > > | CarRun           | 0.97 ± 0.00    | 0.10 ± 0.18   | 118    | 0.98   |
> > > | DroneRun         | 0.29 ± 0.16    | 0.15 ± 0.37   | 117    | 0.98   |
> > > | CarCircle        | 0.69 ± 0.01    | 0.00 ± 0.00   | 120    | 1.00   |
> > > | DroneCircle      | 0.52 ± 0.03    | 0.00 ± 0.00   | 120    | 1.00   |
> > > | CarGoal1         | 0.29 ± 0.05    | 0.87 ± 0.39   | 81     | 0.68   |
> > > | PointCircle1     | 0.52 ± 0.01    | 0.05 ± 0.08   | 118    | 0.98   |
> > > | PointGoal2       | 0.11 ± 0.05    | 0.74 ± 0.33   | 90     | 0.75   |
> > > | AntVelocity      | 0.99 ± 0.01    | 0.54 ± 0.16   | 113    | 0.94   |
> > > | SwimmerVelocity  | 0.26 ± 0.17    | 0.00 ± 0.00   | 120    | 1.00   |
> > >
> > > Overall, CARL achieves high safety rates across most tasks, and performance remains strong with additional seeds.
> > >
> > > - **Reproducibility:**
> > >
> > > The link appears to be functioning on our side, so the issue may have been due to temporary hosting downtime. We will additionally upload a zipped copy of the code.
> > >
> > > - **Theory & Guarantees:**
> > >
> > > This is an example of a typical gap between theory, which must handle the worst case MDP, and practice. Because of this, theoretical constants in results are typically much larger than what should be used in practice.  Indeed -V_{max} is required for theory, but it can be very pessimistic in practice.  The only way to provide a theoretical guarantee using -R_{max} is to restrict the MDP class.  For example, restricting to sparse goal rewards is one class where -R_{max} is sufficient for a guarantee.
> > >
> > >
> > > - **Deployment Penalty:**
> > >
> > > We clarify that the penalty scale is a training-only parameter. The reward relabeling process (Eq. 5) occurs strictly during the offline training phase to shape the policy. At deployment, the agent acts solely based on the learned policy $\pi(s)$, which does not require access to the reward function, penalty values, or the cost critic.

---

> > > > ### Author Response · Authors · 2025-11-21
> > > >
> > > > - **OPE noise:**
> > > >
> > > > Quantifying the exact misclassification rate relative to the ground truth is computationally expensive, as obtaining the true Qc for every transition in the dataset would require rolling out the policy from each transition forward. Instead of measuring the exact noise levels, we measured the method's sensitivity to noise. As detailed in our robustness ablation (random flipping of safe/unsafe decisions based on Q_c), we found that CARL maintains safe performance even when a significant portion of safety decisions are artificially corrupted.  These results demonstrate that high-precision OPE based classification is not a prerequisite for  CARL’s success.
> > > >
> > > > - **contraction/monotone improvement:**
> > > >
> > > > We do not claim theoretical contraction or monotonic improvement; as stated in the paper, the convergence properties of the $M=K=1$ update are an open problem. We will revise the text to temper our claims, specifically clarifying that CARL offers empirical stability in the offline setting rather than theoretical stability guarantees. The claim of alleviating brittleness is based on the observation that CARL consistently produces safe policies across 16 of 19 tasks, whereas theoretically-sound Lagrangian baselines frequently fail in this specific regime.

---

### Author Response · Authors · 2025-12-04
**Summary of Rebuttal**

Dear Area Chair,

We thank the reviewers for their constructive feedback. During rebuttal, we conducted significant additional experiments (robustness stress-tests, new baselines, and architecture generalizations) to address concerns. We summarize the discussion below to assist your decision.

## 1. Addressing Novelty & The Comparison to CPQ (EjRo)

Reviewer **EjRo** questioned the novelty of CARL relative to CPQ. We have demonstrated that while both use constraint estimates, the mechanism of action is fundamentally different:

- **CPQ uses Value Masking $Q = 0$**
  This creates a blind spot (zero gradient). If the critic erroneously flags a safe action as unsafe early in training, the value is zeroed out, and the policy cannot recover.

- **CARL uses Relabeling $r = -V_{\max}$**
  This creates a repulsive landscape (negative gradient), actively pushing the policy away from unsafe regions while allowing recovery if cost estimates improve.

We implemented “CPQ-style” variants of our method (masking in $Q_r$ and/or policy extraction). These variants failed to recover safe policies (Tables in EjRo rebuttal), demonstrating that CARL’s performance gains are driven by the relabeling mechanism, not just the architecture or the strength of the backbone offline RL algorithm.
## 2. Technical Correctness: IQL Integration (EjRo)

Reviewer **EjRo** expressed concern that using an extracted policy breaks the “implicit” nature of IQL. We clarified this misconception:

- **Decoupled Critics:**
  CARL does *not* use the extracted policy to bootstrap the reward value function. The IQL backbone continues to learn rewards via expectile regression on in-sample data, preserving its mathematical properties.

- **Policy Extraction:**
  The explicit policy is extracted only to evaluate the cost critic $Q_c$ for relabeling. This decoupling allows CARL to enforce safety via the penalty term without introducing out-of-distribution (OOD) actions into the IQL value update.
## 3. The Theory–Practice Gap: $V_{\max}$ vs. $R_{\max}$ (tdbs)

Reviewer **tdbs** maintained their score based on the discrepancy between the theoretical requirement for penalty magnitude $V_{\max}\$ and the practical implementation $R_{\max}$.

We acknowledge this gap but characterize it as a *design choice*:

- Using the theoretical $V_{\max}$ results in penalties so large they drown out the reward signal, causing the agent to freeze (e.g., AntVelocity ablation where mean reward dropped from 0.99 to 0.38).

- **Trade-off:**  We trade a loose theoretical bound for **state-of-the-art empirical and robust performance**.
While Lagrangian methods (COptiDICE, BCQ-Lag) possess strict theoretical guarantees, they failed on **17/19 tasks** due to oscillation. CARL, using the $R_{\max}$ penalty, succeeded on **16/19**. We urge the AC to weigh practical utility and stability against theoretical worst-case guarantees.
## 4. Robustness to OPE Error (Reviewers LTcd & chYS)

A primary concern was whether CARL is brittle to errors in the cost critic $Q_c$.

- **New Experiment:**
  We performed a stress test by randomly flipping the safety labels derived from $Q_c$ with probability $p$.

- **Result:**
  CARL maintained safety constraints even with **20–30% label noise**.
  Safety collapsed only when noise reached **50%** (effectively random labels).

These results demonstrate that CARL does not require precise pointwise accuracy; the aggregate topological signal provided by the penalty is sufficient to guide the policy.
## 5. Generalization to New Architectures (chYS)

To show that CARL is not limited to value-based methods, we implemented **CARL-DT (Decision Transformer)**.

- By relabeling the dataset with a learned critic and training a transformer via supervised learning, we satisfied constraints on **6/6** evaluated tasks.

- This demonstrates that CARL acts as a **universal data-wrapper**, compatible with both value-based and sequence-modeling backbones.
## 6. More Results

We provided extensive additional results to contextualize CARL’s performance:

- **Lagrangian Failures:** Lagrangian baselines (BEAR-Lag, BCQ-Lag) failed on **17/19 tasks**. CARL succeeded on **16/19**.
- **New Baselines:** VOCE and WSAC were evaluated and both failed to satisfy strict safety constraints ($\epsilon < 1$) in the tight-budget regime.
- **Statistical Confidence:** We doubled our seed count (3 → 6) for key experiments, with consistent results, and added the *safe trajectory ratio* metric.
# Conclusion

CARL proposes a minimalist, sufficient-condition approach to OSRL. By solving a stricter pointwise proxy problem, we bypass the notorious instability of Lagrangian multipliers in the offline setting. The method is empirically superior to SOTA baselines (FISOR, CAPS), robust to significant OPE noise, and agnostic to the underlying RL backbone.

We hope the extensive empirical validation further reinforces the effectiveness of the approach.

---

### Meta-Review · Area_Chair_P228 · 2026-01-06

**Summary:**

Overall, this paper proposes CARL, a new method for safe reinforcement learning based on a reward relabeling strategy. The main concern raised by the reviewers is the lack of a substantial distinction between CARL and several existing methods, such as CPQ. In addition, some reviewers expressed concerns about the absence of strong theoretical guarantees, although this appears to be a relatively minor issue. Another major concern is the lack of a detailed discussion on how the method addresses the off-policy evaluation (OPE) problem.

**Reviewer Concerns:**

For simplicity, I list only the concerns that I believe have not been fully addressed by the authors.

**Reviewer LTcd:** feasibility under limited offline coverage, conditions of the theoretical results, and the off-policy evaluation issue related to the “no extra hyperparameters” claim.

**Reviewer chYS:** off-policy evaluation (OPE) issue, lack of theoretical guarantees, absence of recent baseline comparisons, and insufficient analysis of cases where CARL occasionally fails.

**Reviewer EjRo:** distinction between CARL and prior methods such as CPQ, and how IQL can be integrated into CARL.

**Reviewer tdbs:** lack of theoretical guarantees.

**Reviewer Scores:**

**Reviewer LTcd:** 4 → 4

**Reviewer chYS:** 4 → 4

**Reviewer EjRo:** 2 → 2

**Reviewer tdbs:** 4 → 4

---

### Decision · Program_Chairs · 2026-01-26

Reject